# Goal-conditioned Reinforcement Learning with Subgoals Generated from Relabeling

## Abstract

In goal-conditioned reinforcement learning (RL), the primary objective is to develop a goal-conditioned policy capable of reaching diverse desired goals, a process often hindered by sparse reward signals. To address the challenges associated with sparse rewards, existing approaches frequently employ hindsight relabeling, substituting original goals with achieved goals. However, these methods exhibit a tendency to prioritize the optimization of closer achieved goals during training, leading to the loss of potentially valuable information from the trajectory and low sample efficiency. Our key insight is that achieved goals, derived from hindsight relabeling, can serve as effective subgoals to facilitate the learning of policies that can reach long-horizon desired goals within the same trajectory. By leveraging these subgoals, we aim to incorporate more longer trajectory information within the same hindsight framework. From this perspective, we propose a novel framework called Goal-Conditioned reinforcement learning with Q-BC (i.e, behavior cloning (BC)-regularized Q) and Subgoals (GCQS) for goal-conditioned RL. GCQS is a innovative goal-conditioned actor-critic framework that systematically exploits more trajectory information to improve policy learning and sample efficiency. As an extension of the traditional goal-conditioned actor-critic framework, GCQS further exploits longer trajectory information, treating them as subgoals that guide the learning process and improve the accuracy of action predictions. Experimental results in simulated robotic environments demonstrate that GCQS markedly improves sample efficiency and overall performance when compared to existing goal-conditioned methods. Additionally, GCQS demonstrated competitive performance on long-horizon AntMaze tasks, achieving results comparable to such state-of-the-art subgoal-based methods.

## 1 Introduction

The integration of Reinforcement Learning (RL) and Deep Learning (DL) has resulted in remarkable progress across various domains. These include advanced robotic control (Quiroga et al., 2022; Qi et al., 2023; Plasencia-Salgueiro, 2023; Zheng et al., 2024), mastery in computer gaming (Quiroga et al., 2022; Zhang et al., 2023a; Plasencia-Salgueiro, 2023; Roayaei Ardakany & Afroughrh, 2024), and sophisticated language processing capabilities (Akakzia et al., 2020; Sharifani & Amini, 2023; Uc-Cetina et al., 2023; Shinn et al., 2024). A critical challenge in RL is fostering efficient learning in scenarios characterized by sparse rewards, a difficulty that is magnified in goal-conditioned RL, thereby adversely affecting sample efficiency. To tackle this issue, Andrychowicz et al. (2017) proposed hindsight experience replay (HER), an approach aimed at significantly enhancing sample efficiency in goal-conditioned RL. HER leverages the abundant repository of failed experiences by relabeling the desired goals in training trajectories with the achieved goals that were actually reached during these failed attempts. This method effectively maximizes the utility of the data available, promoting a more efficient learning process.

HER offers a practical principle for generating pseudo demonstrations to train control policies. Based on HER, several efficient goal-conditioned methods have been proposed, including goal-conditioned actor-critic (GCAC) (Andrychowicz et al., 2017; Fang et al., 2019; Yang et al., 2021) and goal-conditioned weighted supervised learning (GCWSL) methods (Yang et al., 2022; Ma et al., 2022;

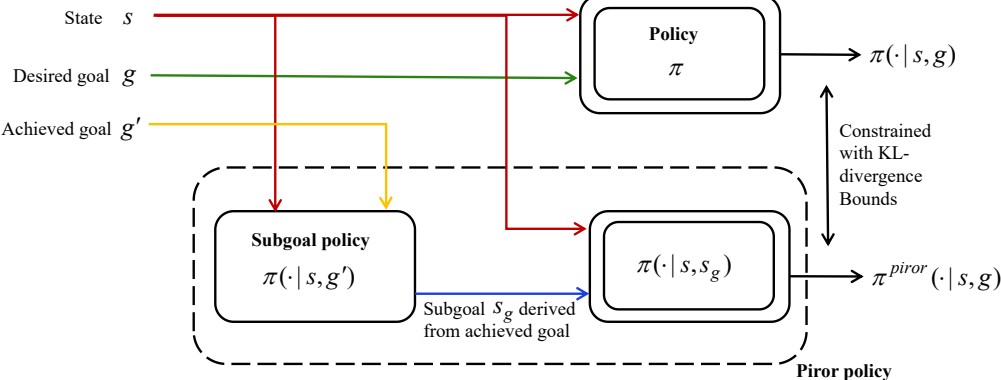

Figure 1: GCQS framework with phasic goal structure in goal-conditioned RL. During training, the policy $\pi$ is constrained to remain close to the prior policy $\pi^{piror}$ through KL-regularization. The prior policy $\pi^{piror}$ is defined as the as the distribution of actions required to reach intermediate subgoals $s_g$ of the task. Notably, the subgoal policy and subgoals are only employed during the training of the target policy $\pi$. At test time, the trained policy $\pi$ is used directly to generate appropriate actions.

Hejna et al., 2023). GCAC focuses on maximizing the Q-function through Temporal Difference (TD)-learning, whereas GCWSL employs weighted behavior cloning.

Despite their success in effectively learning from sparse rewards across various goal-reaching tasks, as we find, both GCAC and GCWSL often exhibit a bias towards sampling short-horizon achieved goals generated from relabeling during policy updates. This bias may lead to suboptimal actions for desired goals that require longer horizons to reach.

From this perspective, we introduce a novel goal-conditioned actor-critic framework, GCQS, designed to enhance action prediction accuracy and further exploit the longer information within the same trajectory. GCQS initially optimizes a Q-BC (i.e, behavior cloning (BC)-regularized Q) objective to efficiently learn to reach achieved (relabeled) goals, similar to the approach employed by GCAC. And then, it utilizes longer achieved goals as subgoals to refine and improve the policy for attaining the desired goals. Specifically, to incorporate subgoals into policy learning, we propose a prior policy within the GCQS framework, which is defined as a distribution over the actions needed to achieve intermediate subgoals (refer to Fig. 3). In light of the results from Paster et al. (2020) and Eysenbach et al. (2022), which demonstrate that imitation learning employed in GCWSL can produce suboptimal policies when dealing with relabeled suboptimal data, we optimize a Q-function objective regularized by behavior cloning (Q-BC) to generate an optimal policy for reaching these subgoals. The prior policy serves as an initial approximation for reaching the desired goals when subgoals are introduced. To refine this process, we implement a policy iteration framework, augmented with a Kullback-Leibler (KL) divergence constraint, specifically designed to guide the refinement of the prior policy (see Fig. 1). We refer to this as a phasic goal structure. To evaluate GCQS, we conduct experiments in standard goal-conditioned gym robotics environments. The experimental results demonstrate that GCQS obtains superior performance and sample efficiency compared to previous goal-conditioned methods, including DDPG+HER (Andrychowicz et al., 2017), Model-based HER (Yang et al., 2021), and various GCWSL approaches (Chane-Sane et al., 2021; Yang et al., 2022; Ma et al., 2022; Hejna et al., 2023). Additionally, GCQS outperforms several advanced subgoal-based algorithms on complex AntMaze tasks. The overall framework of GCQS is shown in Fig. 1.

We briefly summarize our contributions as follows: (1) We demonstrate that both the GCAC and GCWSL methodologies exhibit a tendency to prioritize the learning of actions associated with short-horizon achieved goals, as relabeled from the replay buffer. (2) We propose GCQS, a subgoal-based extension of the GCAC that incorporates longer trajectory information within the hindsight relabeling framework to enhance policy learning efficiency and performance. To the best of our knowledge, GCQS is the first approach to leverage relabeled goals as subgoals to enhance the performance of goal-conditioned policies. Additionally, we provide a detailed analysis demonstrating that this phasic policy structure more accurately predicts actions required to reach desired goals compared to the conventional flat policy structure. (3) Experimental evidence reveals that GCQS outperforms

GCAC and GCWSL in terms of both performance and sample efficiency across various complex goal-conditioned tasks. On more complex long-horizon AntMaze tasks, GCQS achieved performance comparable to such state-of-the-art subgoal-based methods.

## 2 RELATED WORK

**Goal-conditioned Methods**    Addressing goal-conditioned RL tasks involves significant complexities due to the requirement for agents to reach multiple goals concurrently. The major challenge in goal-conditioned RL is managing sparse rewards. To address the issue of sparse rewards, the concept of hindsight was developed, which reinterprets past failures as successes. HER (Andrychowicz et al., 2017) integrates off-policy learning by incorporating hindsight transitions into the replay buffer. This approach enables agents to learn from their experiences by relabeling the goals they initially aimed for with they actually reached (achieved goals). Based on HER, curriculum hindsight experience replay (CHER) (Fang et al., 2019) and model-based hindsight experience replay (MHER) (Yang et al., 2021) introduce heuristically goal selection from failed attempts and model-based goal relabeling, respectively. Goal-conditioned weighted supervised learning (GCWSL) methods (Chane-Sane et al., 2021; Yang et al., 2022; Ma et al., 2022; Hejna et al., 2023) provide theoretical guarantees that learning from achieved goals (relabeled goals) optimizes a lower bound on the goal-conditioned RL objective. In contrast to these methods, GCQS aims to obtain optimal policies to reach these achieved goals by employing a Q-BC objective. This method integrates reinforcement learning and imitation learning, accelerating the learning process. Experimental results demonstrate its superior sample efficiency and performance compared to previous goal-conditioned methods.

**Subgoal Based Approaches**    Several previous studies have suggested employing subgoals to tackle goal-reaching tasks (Jurgenson et al., 2020; Chane-Sane et al., 2021; Kim et al., 2021; Islam et al., 2022; Lee et al., 2022; Zhang et al., 2023b; Kim et al., 2023; Yoon et al., 2024). Our approach diverges from these hierarchical RL methods in that it does not require additional algorithms for subgoal discovery. The closest related work is by Chane-Sane et al. (2021). However, there are significant differences between their method and our GCQS framework. Firstly, Chane-Sane et al. (2021) assumes that the state and goal are identical, which is not applicable in our general goal-conditioned RL environments where states and goals are distinct. Secondly, our method utilizes the relabeled goals within a goal-conditioned RL setting as natural subgoals, thus eliminating the need for separate subgoal discovery mechanisms. This approach has been validated through extensive experimental evaluations. Moreover, Chane-Sane et al. (2021) lacks a theoretical framework explaining why subgoals can enhance policy performance. In contrast, our approach systematically integrates subgoals into the learning process, demonstrating through empirical evidence how these subgoals contribute to improved policy efficiency and effectiveness in realizing possible long-horizon tasks.

## 3 PRELIMINARIES

### 3.1 GOAL-CONDITIONED RL AND HINDSIGHT EXPERIENCE REPLAY

Goal-conditioned reinforcement learning (RL) can be characterized by the tuple $\langle \mathcal{S}, \mathcal{A}, \mathcal{G}, \mathcal{P}, r, \gamma, \rho_0, T \rangle$, where $\mathcal{S}$, $\mathcal{A}$, $\mathcal{G}$, $\gamma$, $\rho_0$ and $T$ respectively represent the state space, action space, goal space, discounted factor, the distribution of initial states and the horizon of the episode. $\mathcal{P} : \mathcal{P}(s'|s, a)$ is the dynamic transition function, and $r : r(s, a, g)$ is typically a simple unshaped binary signal. A typical sparse reward function employed in goal-conditioned RL can be expressed as follows:

$$r(s_t, a_t, g) = \begin{cases} 0, & ||\phi(s_t) - g||_2 < \mu \\ -1, & \text{otherwise} \end{cases} , \tag{1}$$

where $\phi(s_t)$ is the achieved goals, $\mu$ is a threshold and $\phi : \mathcal{S} \to \mathcal{G}$ is a known state-to-goal mapping function from states to achieved goals. HER (Andrychowicz et al., 2017) is an innovative technique designed to enhance learning from unsuccessful attempts and to address the problem of sparse rewards in goal-conditioned RL. HER incorporates four distinct replay strategies to improve the learning process: (1) Final: Replaying transitions corresponding to the final achieved goals of an episode. (2) Future: Replaying transitions with random future achieved goals from the same episode

as the transition being replayed. (3) Episode: Replaying transitions with random achieved goals from within the same episode. (4) Random: Replaying transitions with random achieved goals encountered throughout the entire training process. Among these strategies, the future scheme is generally preferred for goal replay in practical applications. Therefore most prior works and our framework adopt this future strategy to replace desired goals with achieved goals.

## 3.2 GOAL-CONDITIONED ACTOR-CRITIC (GCAC)

GCAC is an efficient temporal-difference (TD)-based RL family of methods enabling agent learns to reach multiple goals with a goal-conditioned policy in goal-conditioned RL. Formally, the objective of a goal-conditioned policy is to maximize expected discounted return:

$$\mathcal{J}(\pi) = \mathbb{E}_{g \sim \rho_g, \tau \sim d^\pi(.|g)} \left[ \sum_t^T \gamma^t r(s_t, a_t, g) \right] \tag{2}$$

under the distribution

$$d^\pi(\tau|g) = \rho_0(s_0) \prod_t^T \pi(a_t|s_t, g) \mathcal{P}(s_{t+1}|s_t, a_t) \tag{3}$$

induced by the policy $\pi$, the initial state $s_0$ and desired goal distribution $g \sim \rho_g$. The policy $\pi(a|s, g)$ utilized in this study yields a probability distribution over continuous actions $a$, conditioned on the state $s$ and desired goal $g$. Several algorithms fundamentally rely on the effective estimation of the state-action-goal value function $Q^\pi$ and the state-goal value function $V^\pi$, which are mathematically expressed as follows::

$$Q^\pi(s, a, g) = \mathbb{E}_{s_0=s, a_0=a, \tau \sim d^\pi(\cdot|g)} \left[ \sum_t^T \gamma^t r(s_t, a_t, g) \right] \tag{4}$$

and

$$V^\pi(s, g) = \mathbb{E}_{a \sim \pi(\cdot|s, g)} Q^\pi(s, a, g). \tag{5}$$

GCAC aims to approximate the $Q^\pi(s, a, g)$ and develop a goal-conditioned policy $\pi(a|s, g)$ that selects actions to maximize $Q^\pi(s, a, g)$. This is obtained through the use of a function approximator, typically a neural network. The learning process involves an iterative approach where the regression of $Q^\pi(s, a, g)$ alternates with the optimization of $\pi$. During this process, the neural network is trained to predict $Q^\pi(s, a, g)$ while simultaneously optimizing $\pi(a|s, g)$ to choose actions that result in high values as determined by $Q^\pi(s, a, g)$. This iterative process ensures that the policy continuously improves by leveraging the learned value function. GCAC is following the standard off-policy actor-critic paradigm such as DQN (Mnih et al., 2015), DDPG (Silver et al., 2014), TD3 (Fujimoto et al., 2018), and SAC (Haarnoja et al., 2018). To further enhance sampling efficiency in goal-conditioned RL, the GCAC framework is often combined with HER. This combination leverages the on the benefits of both approaches, enabling more efficient learning and improved handling of sparse reward environments in goal-conditioned scenarios. In this paper, GCAC refers to the goal-conditioned actor-critic approach combined with the HER variant.

During training, the value function $Q^\pi$ is updated to minimize the TD error:

$$\mathcal{L}_{TD} = \mathbb{E}_{(s_t, a_t, g', s_{t+1}) \sim B_r} \left[ (r'_t + \gamma \hat{Q}^\pi(s_{t+1}, \pi(s_{t+1}, g'), g') - Q^\pi(s_t, a_t, g'))^2 \right], \tag{6}$$

where $\mathcal{B}_r$ is the data distribution after hindsight relabeling, $g'$ represents the achieved goals from $\mathcal{B}_r$, and $\hat{Q}$ refers to the target network which is slowly updated to stabilize training. The policy $\pi$ is trained with policy gradient on the following objective in GCAC:

$$\mathcal{J}_{GCAC}(\pi) = \mathbb{E}_{(s_t, g') \sim B_r} \left[ Q^\pi(s_t, \pi(s_t, g'), g') \right]. \tag{7}$$

## 3.3 GOAL-CONDITIONED WEIGHTED SUPERVISED LEARNING (GCWSL)

In contrast to GCAC methods, which focus on directly optimizing the discounted cumulative return, GCWSL provides theoretical guarantees that weighted supervised learning from hindsight relabeled

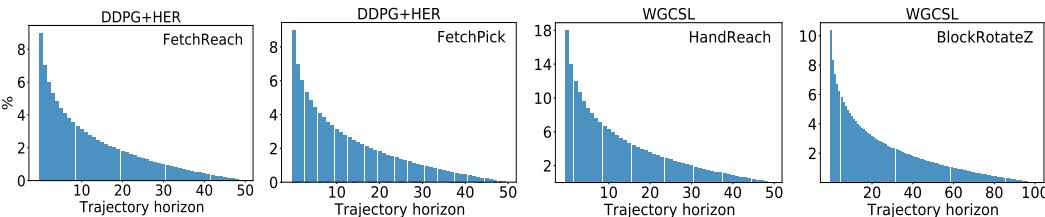

Figure 2: Four example histograms illustrate the distances between initial states and achieved goals when calculating the achieved goals used to update the targets network for DDPG+HER and WGCSL in the Fetch and Hand series tasks. These tasks were trained over a fixed number of epochs: 20 for the Fetch series and 50 for the Hand series. The $X$ axis denotes the horizon between the initial states and achieved goals, while the $Y$ axis represents the percentage of each bin relative to the total updates. This phenomenon suggests a tendency to optimize for shorter distances during the training process, potentially leading to biased learning towards short-horizon goals.

data optimizes a lower bound on the goal-conditioned RL objective. During training, trajectories are sampled form a relabeled dataset by utilizing hindsight mechanisms (Kaelbling, 1993; Andrychowicz et al., 2017). And the policy optimization satisfies the following definition:

$$\mathcal{J}_{GCWSL}(\pi) = \mathbb{E}_{(s_t, a_t, g) \sim \mathcal{D}_r} \left[ w \cdot \log \pi_\theta(a_t|s_t, g) \right], \tag{8}$$

where $\mathcal{D}_r$ denotes relabeled data, $g = \phi(s_i)$ denotes the relabeled goal for $i \geq t$. The weighted function $w$ exists various forms in GCWSL methods (Ghosh et al., 2021; Yang et al., 2022; Ma et al., 2022; Hejna et al., 2023) and can be considered as the scheme choosing optimal path between $s$ and $g$. Therefore GCWSL includes typical two process, acquiring sub-trajectories corresponding to $(s, g)$ pairs and imitating them. In the process of imitation, GCWSL first train the specific weighted function $w$, and then extract the policy with the Equation 10. Note that GCSL (Ghosh et al., 2021) is a special case, and for convenience, we include GCWSL here. Generally, $w \neq 1$.

## 4 GCAC AND GCWSL ARE OFTEN BIASED TOWARDS LEARNING SHORT TRAJECTORIES

The core principle in GCAC and GCWSL is the substitution of desired goals with achieved goals to facilitate the learning process. This strategy leverages the agent's capacity to learn from the states it has successfully reached, thereby promoting effective learning even in the presence of sparse rewards. By focusing on the achieved goals, these frameworks encourage the agent to reinforce its ability to navigate towards goal states it has previously encountered, thus optimizing its policy for a broader range of goal conditions. We use $\tau = \{(s_1, a_1, g, r_1), (s_2, a_2, g, r_2) \ldots, (s_{T-1}, a_{T_{max}-1}, g, r_{T_{max}-1}), s_{T_{max}})\}$ to denote a trajectory visited by state in replay buffer, and $\tau^{g'} = \{\phi(s_1), \phi(s_2), \ldots, \phi(s_{T_{max}-1}), \phi(s_{T_{max}})\}$ denotes the achieved goal trajectory. GCAC and GCWSL alternates $g$ and $r_t$ in the t-th transition $(s_t, a_t, g, r_t, s_{t+1})$ with a future achieved goal $g' = \phi(s_{i+t+1}), 1 \leq i \leq T_{max} - t$ selected from achieved goal trajectory and $r'_t = r(s_{i+t+1}, a_{i+t+1}, g')$ in the same suffix. Upon relabeling, transitions within failed trajectories can be assigned non-negative rewards. Consequently, HER effectively mitigates the primary challenge of sparse rewards in goal-conditioned RL. To be precise, the process involves sampling $t \sim U(1, T_{max} - 1)$ which determines the current state $\tau(s_t)$. Subsequently, an achieved goal is selected from the achieved goal trajectory: $\tau^{g'}(\phi(s_{i+t+1})), i \sim U(1, T_{max} - t)$, where $i$ is the chosen future offset. We define $p(i)$ probability of selecting a future offset with a horizon length $i$. This leads us to establish the following theorem:

**Theorem 4.1.** *The cumulative function $S(x(K)) := \sum_{k \geq K} x_k$ of the probability $p$ of fixed offset horizon length $I$ for GCAC and GCWSL updates is characterized by a monotonically decreasing function:*

$$S(p(I+1)) \leq S(p(I)). \tag{9}$$

The proof is available in Appendix A.1. This theorem remains unaffected by the value of $p(i)$, even though $p(i)$ is derived from the transition dynamics $\mathcal{P}$ and the behavior policy, as demonstrated in Eq. (2) and Eq. (3). Consequently, we infer that within the HER framework, both GCAC and

GCWSL are predisposed to select achieved goals with shorter horizons for relabeling and updating. We performed a statistical analysis on the time step offsets $i$ used for updates in a fixed number of DDPG+HER examples within GCAC and GCWSL, determining the percentage distribution of each time step offset (refer to Fig. 2).

The analysis demonstrates that a significant portion of the updates is concentrated on relatively short segments of sub-trajectories, despite the trajectories often reaching their maximum permissible length, $T_{max}$, illustrated at the furthest right of the $X$ axis. This pattern indicates a pronounced inclination within these methods to favor updates concerning immediate goals, resulting in a model that primarily acquires information from scenarios involving goals with shorter horizons.

## 5 GCQS: AN EXTENDED VERSION OF GCAC

Based on the insights and analysis from Section 4, we have developed a novel framework for goal-conditioned RL called GCQS. The primary motivation behind GCQS is to leverage more extensive long trajectories for updates. Overall of this framework is illustrated in Fig. 1. Since we find that GCWSL underperforms compared to GCAC in our experiments, which may be attributed to GCWSL's lack of stitching capability (Cheikhi & Russo, 2023; Ghugare et al., 2024). Therefore GCQS integrates the SAC following GCAC. The core of GCQS is grounded in the observation that it is generally more straightforward to identify future achieved goals that lead to the ultimate desired goals, rather than determining the optimal action directly from the initial state. By redefining these achieved goals as subgoals and embedding them within GCAC models, the accuracy of action predictions can be significantly enhanced. This process not only simplifies the learning trajectory but also improves the overall efficiency and effectiveness of the policy learning framework.

In the following sections we describe the specific implementation and analysis of GCQS. We first introduce a policy $\pi(\cdot|s, g')$ for reaching achieved goals, as detailed in Section 5.1. Next, we enhance the desired goal-conditioned policy $\pi(\cdot|s, g)$ by using achieved goals trajectory as subgoals distribution, as discussed in Section 5.2.

### 5.1 OBTAIN THE OPTIMAL POLICY TO REACH THE ACHIEVED GOALS VIA Q-BC

In this section, we elucidate the process for training a policy to effectively reach achieved goals, specifically when utilizing the future strategy.

First, we posit the existence of a relabeling policy $\pi_{relabel}$ capable of generating achieved goals $g'$ within the relabeled data $\mathcal{B}_r$. Our goal-conditioned policy that reaching achieved goals is then trained to optimize the following objective function while adhering to KL-divergence constraints:

$$\arg\max_{\pi}\mathbb{E}_{(s,g')\sim\mathcal{B}_r, a\sim\pi(s,g')}[Q^{\pi}(s, a, g')], \text{s.t. } \mathcal{D}_{\text{KL}}\left(\pi\|\pi_{relabel}\right) \leq \epsilon. \tag{10}$$

Since minimizing the KL-divergence corresponds to optimizing for maximum likelihood (LeCun et al., 2015):

$$\min \mathcal{D}_{\text{KL}}\left(\pi\|\pi_{relabel}\right) = \min \mathbb{E}_{\mathcal{B}_r}\left[\log \pi(a|s, g')\right]. \tag{11}$$

and considering a stochastic policy, we have the following Lagrangian equation:

$$\mathcal{L}(\lambda, \pi) = \mathbb{E}_{a\sim\pi(\cdot|s,\phi(s))}\left[Q^{\pi}(s, a, g')\right] + \lambda\mathbb{E}_{(s,a,\phi(s))\sim\mathcal{B}_r} \log \pi(a|s, g').$$

In this case, the stochastic policy $\pi(s, g')$ can be regarded as a Dirac-Delta function thus, the $\int_a \pi(a|s, g')da = 1$ constraint always satisfies. Therefore optimization objective become:

$$\arg\max_{\pi}\mathbb{E}_{(s,a,g')\sim\mathcal{B}_r}\left[Q^{\pi}(s, a, g') + \log(\pi(a|s, g'))\right]. \tag{12}$$

We refer to our goal-conditioned policy objective that reaches achieved goals in Eq. (12) as Q-BC.

**Compared with GCAC** In practice, the Q-BC objective integrates reinforcement learning (by maximizing $Q^{\pi}$) with imitation learning (by maximizing the behavior cloning). This integration effectively accelerates the GCAC learning process through behavior cloning regularization derived from relabeled data. This concept aligns with various methods designed to expedite reinforcement

learning through demonstrations (Atkeson & Schaal, 1997). Historically, behavior cloning has been employed to regularize policy optimization using natural policy gradients (Kakade, 2001; Lillicrap et al., 2015; Rajeswaran et al., 2017; Nair et al., 2018; Goecks et al., 2019), often incorporating additional complexities such as modified replay buffers and pre-training stage. Moreover, our Q-BC approach eliminates the need for additional parameters while maintaining training stability, akin to the methods discussed in Fujimoto & Gu (2021).

## 5.2 Policy Improvement with Subgoals derived from achieved goals

In this section, we redefine the well-learned achieved goals $g'$ as subgoals $s_g$ that facilitate reaching the desired goals $g$. This approach enhances the learning process by integrating intermediate objectives that guide the agent towards its ultimate goal, leveraging the structure provided by the achieved goals to optimize the overall policy. The key perspectives in this section can be visualized in the Fig. 3. To formalize this notion, we first introduce a KL constraint on the policy distribution, conditioning on desired goals $g$ and subgoals $s_g$:

$$\mathcal{D}_{\mathrm{KL}}\left(\pi(\cdot|s,g)||\pi(\cdot|s,s_g)\right) \leq \eta. \tag{13}$$

In goal-conditioned RL, for a given state $s$ and desired goal $g$, we implement a bootstrapping technique to estimate the policy's performance at subgoals $s_g$. These subgoals are sampled from the trajectory distribution of achieved goals $\tau^{g'}$. Then we have the following definition for the prior goal-conditioned policy that reaches desired goals:

$$\pi^{prior}(a|s,g) := \mathbb{E}_{s_g \sim \tau^{g'}}\left[\pi(a|s,s_g)\right]. \tag{14}$$

Given the premise that subgoals are typically more reachable than final desired goals, we utilize the prior policy as a valuable initial estimate to guide the search for optimal actions. To ensure proper alignment of the policy behavior, we introduce a policy iteration framework that incorporates an additional KL divergence constraint. During the policy improvement stage, in addition to maximizing the Q-function as specified in Eq. (12), we integrate a KL regularization term to maintain the policy's proximity to the prior policy. This regularization helps ensure consistency with the initial estimate, thereby facilitating a more efficient search for optimal actions.

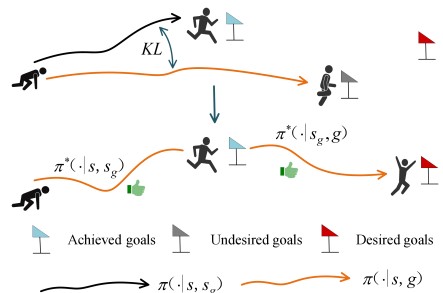

Figure 3: Achieved goals $g'$ are considered subgoals $s_g$ because they are easy to reach and bounding KL-constrained optimal path for reaching $s_g$ and desired goals $g$.

Therefore the desired goal-conditioned policy objective can be expressed as follows:

$$\arg\max_{\pi}\mathbb{E}_{(s,g)\sim\mathcal{B}}\mathbb{E}_{a\sim\pi(\cdot|s,g)}\left[Q^\pi(s,a,g) - \beta\mathcal{D}_{\mathrm{KL}}\left(\pi(\cdot|s,g) \parallel \pi^{prior}(\cdot|s,g)\right)\right], \tag{15}$$

where $\beta$ is a hyperparameter. The construction of prior policy in Eq. (14) and KL-divergence term in Eq. (15) are estimated by Monte-Carlo approximation followed Chane-Sane et al. (2021), ensuring stable convergence. This phasic goal-conditioned policy structure enables the derivation of more optimal actions for potentially long-horizon goals. We will provide practical implementation of the entire algorithm and analyze why this phasic structure is better than the previous flat structure in detail in Appendix B.1.

Although prior work on subgoal policies has rarely provided performance guarantees, we draw upon the insights from Ma et al. (2022) to demonstrate that iterative learning under the structured properties of phasic structure policy can yield statistical guarantees for the optimal policy of GCQS, as described in Eq. (15).

**Theorem 5.1** (Performance Guarantee). *Assume* $\sup|r(s,a,g)| \leq R_{\max}$. *Consider a policy class* $\Pi : \{S \rightarrow \Delta(A)\}$ *such that* $\pi^* \in \Pi$. *Then, for any* $\delta$, *with probability at least* $1 - \delta$, *GCQS framework will return a policy* $\hat{\pi}$ *such that:*

$$\sup_{s,g}\left|V^*(s,g) - V^{\hat{\pi}}(s,g)\right| \leq \frac{R_{\max}\sqrt{2\eta}}{1-\gamma} + \frac{R_{\max}\sqrt{2\log\left(\frac{|\Pi|}{\delta}\right)}}{\sqrt{N}}. \tag{16}$$

The proof is available in Appendix A.2. This theorem provides a theoretical performance guarantee for the GCQS algorithm in goal-conditioned reinforcement learning, explicitly defining the upper bound on the V-value function error between the learned policy $\hat{\pi}$ and the optimal policy $\pi^*$. The theorem demonstrates that the error bound is influenced by the upper bound on KL-divergence $\eta$ and the number of samples $N$. By controlling $\eta$, the policy deviation can be constrained, ensuring stability during policy optimization. Additionally, increasing the sample size improves the approximation accuracy of the policy. While the theorem depends on the quality of the prior policy, it offers a strong theoretical foundation for the practical effectiveness and sample efficiency of the GCQS algorithm.

## 6 EXPERIMENTS

We begin by presenting the benchmarks and baseline methodologies utilized in our study, accompanied by a detailed description of the experimental procedures. Following this, we report the results and provide a thorough analysis, demonstrating how they corroborate our initial assumptions and theoretical framework.

**Benchmarks**   We utilize the established goal-conditioned research benchmarks as detailed by Plappert et al. (2018), encompassing four manipulation tasks on the $Shadow - hand$ and all tasks on the $Fetch$ robot. We also conducted comparisons with an advanced subgoal algorithm on the complex long-horizon AntMaze tasks used in Hu et al. (2023). Fig. 4 presents examples of the tasks.

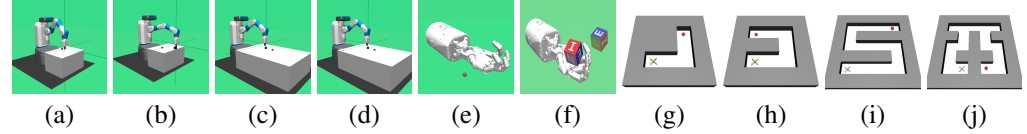

| (a) | (b) | (c) | (d) | (e) | (f) | (g) | (h) | (i) | (j) |

Figure 4: Goal-conditioned example tasks: (a) FetchReach, (b) FetchPush, (c) FetchSlide, (d) FetchPickAndPlace, (e) HandReach, (f) HandManipulateBlock. (h) L-AntMaze. (i) U-AntMaze. (j) S-AntMaze. (g) $\pi$-AntMaze.

**Baselines**   In this section, we conduct a comparative analysis of our proposed method against various established goal-conditioned policy learning algorithms. We implemented the baseline algorithms within the same off-policy actor-critic framework as our method to ensure a consistent and fair evaluation. All experiments are conducted using five random seeds. Detailed algorithm implementation is described in Appendix C. We compare with following goal-conditioned baselines including GCAC and GCWSL methods: (1) **DDPG** (Lillicrap et al., 2015), which is an off-policy actor-critic method for learning continuous actions. (2) **DDPG+HER** (Andrychowicz et al., 2017), which combines DDPG with HER, which learns from failed experiences with sparse rewards. (3) **MHER** (Yang et al., 2021), which constructs a dynamics model using historical trajectories and combines current policy to generate virtual future trajectories for goal relabeling. (4) **GCSL** (Ghosh et al., 2021), which incorporates hindsight relabeling in conjunction with behavior cloning to imitate the suboptimal trajectory. (5) **WGCSL** (Yang et al., 2022) builds upon GCSL by incorporating both goal relabeling and advantage-weighted updates into the policy learning process, and can be applied to both online and offline settings. (6) **GoFar** (Ma et al., 2022) employs advantage-weighted regression with $f$-divergence regularization based on state-occupancy matching. (7) **DWSL** (Hejna et al., 2023), which initially creates a model to quantify the distance between given state and the goal and policy derivation involves imitating actions that effectively minimize this distance metric. We also performed comparisons with state-of-the-art subgoal-based methods on complex AntMaze tasks, as described in Yoon et al. (2024). These methods include BEAG (Yoon et al., 2024), PIG (Hu et al., 2023), DHRL (Lee et al., 2022), and HIGL (Kim et al., 2021).

### 6.1 PERFORMANCE EVALUATION ON GOAL-CONDITIONED BENCHMARKS RESULTS

For all experiments, we use a single GPU to train the agent for 20 epochs in Fetch tasks and 50 epochs in Hand tasks. Upon completing the training stage, the most effective policy is evaluated by testing it on the designated tasks. The performance outcomes are then expressed as mean success

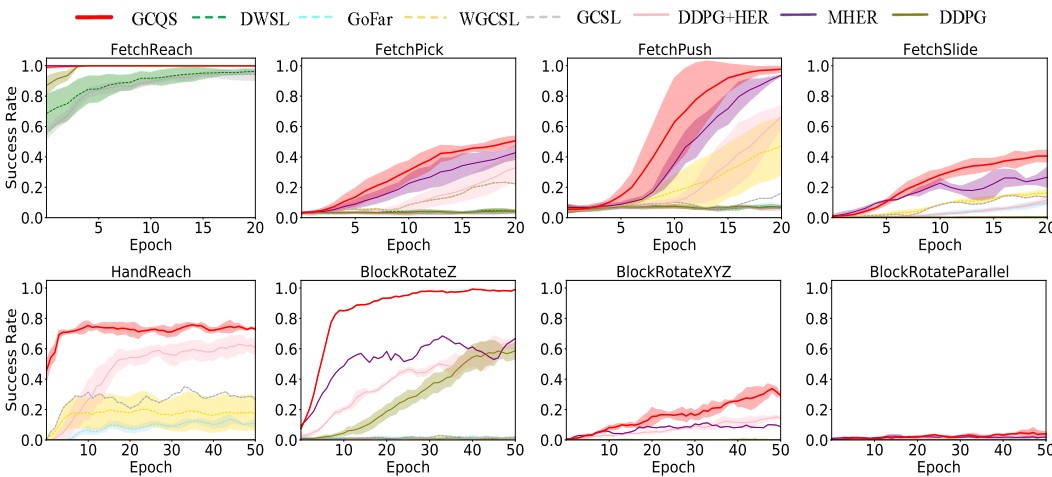

Figure 5: Performance on eight robot goal-reaching tasks in goal-conditioned benchmarks. Results are averaged over five random seeds and the shaded region represents the standard deviation.

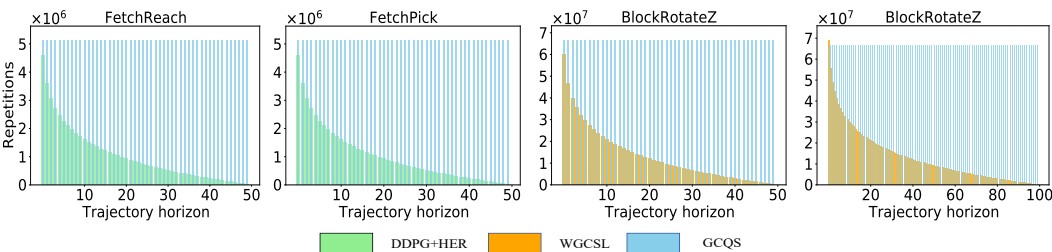

Figure 6: Histogram of lengths of successful trajectories in the four goal-conditioned tasks. X axis is the length of the successful trajectory, Y axis is the bin count for that length. The histograms show that GCQS successes are more concentrated on long trajectories compared to DDPG+HER and WGCSL.

rate. Performance comparisons across training epochs are illustrated in Fig. 5. As illustrated in Fig. 5, GCQS demonstrates significantly superior performance compared to the other baseline methods, coupled with a markedly faster learning speed. The results indicate that DDPG and Actionable Models exhibit slow learning across all tasks, whereas other methods benefit from HER, showcasing its critical role in enhancing learning efficiency and handling sparse rewards in goal-conditioned RL.

Interestingly, the advanced algorithms DWSL and GoFar perform poorly, likely due to their configurations being more suited for offline goal-conditioned RL. Furthermore, we compared our method with two representative approaches, DDPG+HER and WGCSL, during the update process, as shown in Fig. 6. It is evident that GCQS effectively addresses the issue of short trajectory updates, applying robustly across all trajectory lengths, especially for longer trajectories.

## 6.2 Performance Evaluation on complex Antmaze Results

As illustrated in Fig. 7, although GCQS does not incorporate additional algorithms to determine subgoal selection, it demonstrates performance comparable to the advanced SOTA algorithms on L-Antmaze task. This indicates that selecting subgoals from relabeled data is highly effective. In the U-Antmaze, S-Antmaze, and $\pi$-Antmaze environments, GCQS demonstrates performance slightly inferior to or comparable with PIG, but outperforms both HIGL and DHRL. Further research could focus on refining methods for choosing more suitable subgoals from the relabeled goals.

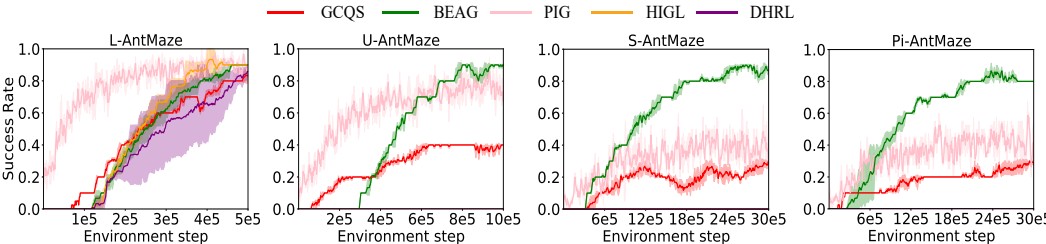

Figure 7: Performance on four complex long-horizon Antmaze tasks. We note that certain baselines may not be visible in specific environments due to overlapping values, especially at zero success rates.

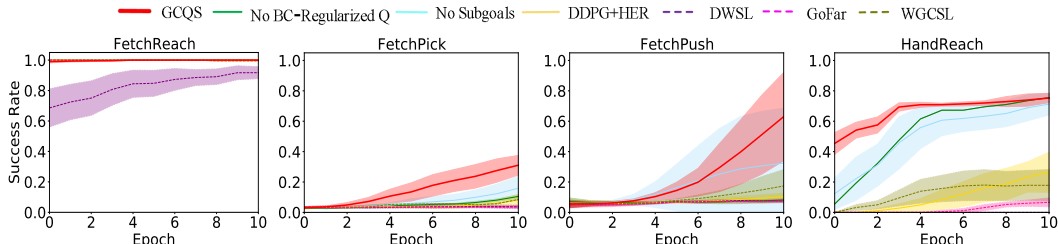

Figure 8: Ablation studies in FetchReach, FetchPick, FetchPush and HandReach.

## 6.3 ABLATION STUDIES

To evaluate the significance of subgoals and BC regularization during the stage of learning achieved goals in the GCQS framework, we conducted a series of ablation experiments comparing GCQS variants with HER. In these experiments, the number of subgoals corresponds to all achieved goals, and the parameter $\beta$ is set to 0.2 by default. We experiment with the following settings:

- **GCQS** SAC+Q-BC+Subgoals.
- **No BC-Regularized Q** which is equivalent to remove KL constraints.
- **No Subgoals** which is equivalent to apply flat goal-conditioned policy.

The empirical results shown in Fig. 8 demonstrate that subgoals are more pivotal than BC-Regularized Q within the GCQS framework. The GCQS method attains faster learning compared to competitive baseline DDPG+HER, while the state-of-the-art DWSL struggles to learn effectively in these tasks, with the exception of FetchReach. This observation implies that supervised learning (SL) approaches are suboptimal for relabeled data.

Integrating BC-Regularized Q with subgoals leads to substantial performance enhancements. This improvement arises from the synergistic interaction between BC-Regularized Q and subgoals within the GCQS framework. Subgoals offer an improved policy for attaining desired goals, while BC-Regularized Q fine-tunes this policy, thereby efficiently directing the subgoal curriculum.

## 7 CONCLUSION

This paper presents GCQS, an advanced GCAC framework for goal-conditioned RL that incorporates a subgoal generation strategy. This approach is motivated by the observation that existing goal-conditioned methods tend to prioritize updates on short-horizon trajectories. A distinctive feature of GCQS is its ability to autonomously generate subgoals using the same relabeling technique applied to the same trajectory, thereby removing the need for additional discovery mechanisms. By leveraging longer trajectories as intermediate subgoals, GCQS enhances the agent's capacity to predict more accurate actions. Future work will focus on developing more refined techniques for identifying subgoals from accomplished outcomes, further optimizing the training of goal-conditioned policies.

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

# Contents of Appendix

# A PROOFS

In this section, we restate theorems in the paper and present their proofs.

## A.1 PROOF OF THEOREM 4.1

First we write $S(p(I))$ as:

$$S\big(p(I)\big) = \sum_{i \geq I} p_i = \sum_{i \geq I} p_i + \sum_{i < I} p_i \cdot 0. \tag{17}$$

Then we can obtain $S(p(I+1))$ as:

$$S\big(p(I+1)\big) = \sum_{i \geq I+1} p_i + \sum_{i < I+1} p_i \cdot 0. \tag{18}$$

Comparing Eq. (17) and Eq. (18), we can see that:

$$S\big(p(I)\big) = p(I) + S\big(p(I+1)\big), \tag{19}$$

since $p(I) \geq 0$, we have that the cumulative function $S$ of the probability of fixed offset horizon length $I$ is monotonically decreasing:

$$S\big(p(I+1)\big) \leq S\big(p(I)\big). \tag{20}$$

## A.2 PROOF OF THEOREM 5.1

**Notation**. Let $\pi^*$ be the optimal policy and $\hat{\pi}$ be the policy returned by GCQS. The $\pi^*$ satisfies $V^*(s,g) = \max_{\hat{\pi}} V^{\hat{\pi}}(s,g)$. $\gamma \in (0,1)$ is the discount factor.

**Assumptions**. Before proving this theorem, we first have the following assumptions:

1. For all states $s$, actions $a$, and goals $g$, the reward function satisfies $|r(s,a,g)| \leq R_{\max}$.
2. The size of the policy class $\Pi$ is $|\Pi|$ and $\delta$ represents the confidence level controlling the error bound.
3. The training samples $a_1, a_2, \ldots, a_N$ are independently and identically distributed (IID) from the policy .

**Proof**. Since our algorithm is built on the basis of GCAC, we can define the error between $V^*(s,g)$ and $V^{\hat{\pi}}(s,g)$ as:

$$
\begin{aligned}
\delta V(s,g) =: & V^*(s,g) - V^{\hat{\pi}}(s,g) \\
= & \max_a \left[ r(s,a,g) + \gamma \mathbb{E}_{s' \sim \mathcal{P}} \left[ V^*(s',g) \right] \right] - \mathbb{E}_{a \sim \hat{\pi}} \left[ r(s,a,g) + \gamma \mathbb{E}_{s' \sim \mathcal{P}} \left[ V^{\hat{\pi}}(s',g) \right] \right] \\
= & \gamma \mathbb{E}_{s' \sim \mathcal{P}} \left[ V^*(s',g) - V^{\hat{\pi}}(s',g) \right] + \left( \max_a r(s,a,g) - \mathbb{E}_{a \sim \hat{\pi}} [r(s,a,g)] \right)
\end{aligned}
\tag{21}
$$

Since $\delta V(s,g)$ reflects the difference between the policies $\hat{\pi}$ and $\pi^*$, we need to quantify this difference further. In GCQS, the policy $\hat{\pi}$ satisfies a KL-divergence constraint with respect to the prior policy $\pi^{piror}$:

$$D_{\mathrm{KL}}(\hat{\pi}(\cdot|s,g) \| \pi_{\mathrm{prior}}(\cdot|s,g)) \leq \eta, \quad \forall s, g. \tag{22}$$

Using Pinsker's Inequality (Pinsker, 1964), we can obtain:

$$\|\hat{\pi}(\cdot|s,g) - \pi^{prior}(\cdot|s,g)\|_1 \leq \sqrt{2 D_{\mathrm{KL}}(\hat{\pi}(\cdot|s,g) \| \pi^{prior}(\cdot|s,g))} \leq \sqrt{2\eta}. \tag{23}$$

We consider the impact of policy differences on the V-value function error. Using the recursive nature of Bellman error and the maximum difference impact, we have:

$$
\begin{aligned}
|\delta V(s,g)| \leq & \left| \max_a r(s,a,g) - \mathbb{E}_{a \sim \hat{\pi}} [r(s,a,g)] \right| + \gamma \mathbb{E}_{s' \sim \mathcal{P}} \left[ |V^*(s',g) - V^{\hat{\pi}}(s',g)| \right] \\
\leq & R_{\max} \|\pi^*(\cdot|s,g) - \pi_{\hat{\theta}}(\cdot|s,g)\|_1 + \gamma \mathbb{E}_{s' \sim \mathcal{P}} \left[ |V^*(s',g) - V^{\hat{\pi}}(s',g)| \right] \\
\leq & R_{\max} \sqrt{2\eta} + \gamma \mathbb{E}_{s' \sim \mathcal{P}} \left[ |V^*(s',g) - V^{\hat{\pi}}(s',g)| \right] \\
\leq & \frac{R_{\max} \sqrt{2\eta}}{1-\gamma}
\end{aligned}
\tag{24}
$$

The first line utilizes the inequality $\mathbb{E}[|X|] \geq |\mathbb{E}[X]|$. The second line shows that the discrepancy in immediate rewards can be controlled through the distribution of action selection. The fourth line is derived through the recursive expansion of the future value differences. Additionally, considering the effect of sample size $N$ on policy learning, we use Hoeffding's inequality (Hoeffding, 1994) to further limit the value function estimation error under finite samples. Here's the detailed process. First, let us review Hoeffding's inequality. Hoeffding's inequality is a concentration inequality that provides a bound on the deviation of the sum of bounded independent random variables. For given random variables $(X_1, X_2, \ldots, X_N)$ bounded within an interval $[a, b]$, the probability of deviation from the expected value can be bounded as follows:

$$P\left(\left|\frac{1}{N}\sum_{i=1}^{N} X_i - \mathbb{E}[X]\right| \geq \epsilon\right) \leq 2\exp\left(-\frac{2N\epsilon^2}{(b-a)^2}\right) \tag{25}$$

Equivalently, for a given confidence level $1 - \delta$, the inequality can be inverted to yield an upper bound on the deviation:

$$\left|\frac{1}{N}\sum_{i=1}^{N} X_i - \mathbb{E}[X]\right| \leq \sqrt{\frac{(b-a)^2 \log(2/\delta)}{2N}} \tag{26}$$

Then, we apply it to value function estimation. To apply Hoeffding's inequality within our setting, we assume that we have $N$ independent samples $(s, a, g)$ for estimating the value function $V^\pi(s, g)$. Given that the reward function $r(s, a, g)$ is bounded by $|r(s, a, g)| \leq R_{\max}$, the discrepancy between the empirical and true values of $V^\pi(s, g)$ can be controlled using Hoeffding's inequality. Specifically, we obtain the following bound on the error of the value function estimation with confidence $1 - \delta$:

$$\sup_{s,g}|V^*(s, g) - V^\pi(s, g)| \leq R_{\max}\sqrt{\frac{2\log(|\Pi|/\delta)}{N}} \tag{27}$$

Finally, combining the infinite-sample bound from Eq. (24) with the finite-sample bound derived via Hoeffding's inequality, we arrive at the following refined bound:

$$\sup_{s,g}\left|V^*(s, g) - V^{\hat{\pi}}(s, g)\right| \leq \frac{R_{\max}\sqrt{2\eta}}{1-\gamma} + \frac{R_{\max}\sqrt{2\log\left(\frac{|\Pi|}{\delta}\right)}}{\sqrt{N}}. \tag{28}$$

## B  GCQS TECHNICAL DETAILS

In this section, we provide additional technical details of GCQS that are omitted in the main text. These include (1) detail of the overall GCQS algorithm, and (2) phasic policy structure analysis in GCQS.

### B.1  PRACTICAL GCQS ALGORITHM

The complete GCQS algorithm is detailed in Algorithm 1. GCQS extends the SAC framework within GCAC. For each episode, a goal $g$ is sampled from the desired goal distribution, and a trajectory is collected using the current policy as behavior policy. This trajectory is subsequently stored in the replay buffer $\mathcal{B}$. Following data collection, a minibatch $m$ is sampled from the replay buffer. The future strategy is employed to relabel goals in the minibatch with achieved goals $g' = \phi(s_i)$. After hindsight relabeling, the minibatch $m$ belongs to the relabeled distribution $\mathcal{B}_r$ and is used to train both the $Q^\pi$ network and the policy network. The $Q^\pi$ network is updated according to Eq. (6), and the subgoal policy is trained to minimize the Q-BC objective as described in Eq. (12). Finally, these achieved goals are reused as subgoals to refine the policy by maximizing the KL-divergence regularized $Q^\pi$ function as described in Eq. (15).

### B.2  PHASIC POLICY STRUCTURE ANALYSIS

To further elucidate the advantages of our phasic goal-conditioned policy structure in Section 5.2, we analyze an example trajectory between a randomly selected state and a desired goal $(s, g)$ under

---

**Algorithm 1** GCQS For Goal-conditioned RL

---

1: **Initialize** off-policy replay buffer $\mathcal{B}$
2: **Initialize** policy $\pi_\theta$, policy target network weights $\bar\theta \leftarrow \theta$, value $Q_\psi$, value target network weights $\bar\psi \leftarrow \psi$
3: **while** a fixed number of iteration **do**
4:     Sample goal from desired goal distribution $g \sim p(g)$
5:     Collect trajectories with the policy $\pi$ and save to the replay buffer $\mathcal{B}$
6:     **for** Update goal-conditioned policy step **do**
7:         Sample a minibatch $m$ from the replay buffer: $\{(s_t, a_t, g, r_t)\} \sim \mathcal{B}$
8:         Relabel $g$ with achieved goals through future strategy for minibatch $m$: $m \leftarrow \{(s_t, a_t, \phi(s_i), r(s_t, a_t, \phi(s_i))), i \geq t\}$, where $\phi(s_i)$ is the achieved goals
9:         Compute target $y_t$ with $y_t = r'_t + \gamma \hat{Q}_{\bar\psi}(s_{t+1}, \pi(s_{t+1}, g'), g')$ and minimize critic loss in Eq. (6)
10:         //Subgoal Policy Learning
11:         Update $\pi_\theta^{pior}$ with minimize actor loss in Eq. (12)
12:         //Policy Improvement with Subgoals Derived from Achieved Goals
13:         Update $\pi_\theta$ with minimize actor loss in Eq. (15)
14:     Soft update the target policy and value network: $\bar\theta \leftarrow \tau\theta + (1-\tau)\bar\theta, \bar\psi \leftarrow \tau\psi + (1-\tau)\bar\psi$

---

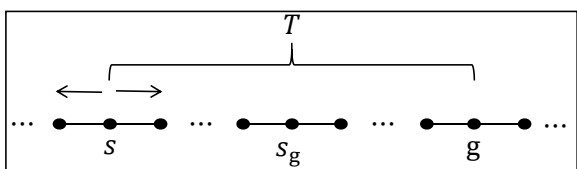

Figure 9: One-dimensional state space and goal space trajectory example between $s$ and $g$. In this trajectory, the agent can only perform left or right actions at each time step with equal transition probability. Similar to reward definition in Eq. (1), the agent gets a reward of 0 when it reaches the desired goal and -1 in otherwise. We assume that $T$ is the horizon distance between state $s$ and desired goal $g$ which satisfies $g = s + T$, $i$ is the horizon distance between state $s$ and subgoal $s_g$.

certain assumptions motivated by Park et al. (2024), as illustrated in Fig. 9. This example demonstrates the intermediate stages and decision points within the trajectory, highlighting the effectiveness of integrating subgoals into the learning process. Through this analysis, we aim to provide a clearer understanding of how phasic structure enhances the policy's ability to navigate towards long-horizon goals while maintaining adaptability and robustness.

Based on the above description the optimal goal-conditioned $Q$-value function is hence given as

$$Q^* = -T - 1. \tag{29}$$

**Proof**. In the one-dimensional state space and goal space example in Fig. 9, we obviously obtain the optimal $V$-value is $V^*(s, g) = -T$ (assume that $\gamma = 1$). According to $Q^\pi(s, a, g) = r(s, a, g) + \gamma \sum_{s' \in S} \mathcal{P}(s'|s, a) V^\pi(s', g)$ we can obtain:

$$\begin{aligned} Q^*(s, a, g) &= -1 + \mathcal{P}_1 V^*(s-1, g) + \mathcal{P}_2 V^*(s+1, g) \\ &= -1 + \mathcal{P}_1(T-1) + \mathcal{P}_2(T+1) \\ &= -T + \mathcal{P}_2 - \mathcal{P}_1 - 1 \\ &= -T - 1. \end{aligned} \tag{30}$$

GCQS builds upon the GCAC framework by first fitting a $Q^\pi$ function and then extracting a policy that selects actions leading to high-value outcomes. However, when the goal $g$ is distant from the current state $s$, the goal-conditioned value function may struggle to provide a clear learning signal for a straightforward goal-conditioned policy. This issue arises for two main reasons:

The first is the precise estimation of the value function. As the distance between $s$ and $g$ increases, the precision of the $Q^\pi$ values tends to decrease. This reduction in precision occurs because the differences in $Q^\pi$ values for subsequent states $Q^\pi(s_{t+1}, a, g)$ may be minimal. Consequently, suboptimal actions can be easily corrected within a few steps, resulting in only minor penalties.

The second is the noise and error accumulation. The $Q^\pi$ function's noise and errors, including sampling and approximation errors, become more pronounced when the goal $g$ is far from the current state $s$. These errors can overshadow the minor differences in value estimates, making it challenging for the policy to distinguish between optimal and suboptimal actions effectively. This issue is exacerbated when the magnitude of the value function, and consequently its noise, is large due to the long horizon involved. By addressing these two issues, GCQS aims to provide a more robust and efficient approach to goal-conditioned RL, particularly in scenarios involving long-horizon goals.

We assume that the noise in the learned value function $\hat{Q}^\pi(s, a, g)$ is proportional to the optimal value: i.e., $\hat{Q}^\pi(s, a, g) = Q^*(s, a, g) + \sigma z_{s,g} Q^*(s, a, g)$, where $z_{s,g}$ is sampled independently from the standard normal distribution, and $\sigma$ is its standard deviation. This assumption implies that noise increases as the desired goal becomes more distant. We illustrate this concept with references in Fig. 10, where the distance represents the horizon length between the state $s$ and the desired goal $g$. The curve illustrates a clear trend: as the distance between the state and the goal increases, the learned value function exhibits greater noise.

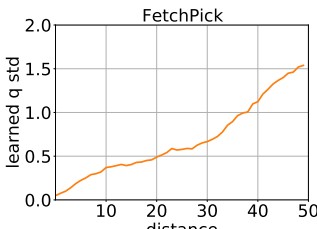

Figure 10: The relationship curve between distance and standard deviation in the FetchPick Task.

In this case, we assess the probability of selecting incorrect actions when comparing flat and phasic goal-conditioned policies. Note that we define achieved goal policy as subgoal policy. The subgoal policy evaluates values at $s \pm i$ by considering subgoals that are $i$-steps away. For the phasic structure approach, both the subgoal and desired goal policies are queried at each step. This methodology allows us to derive the bounds on the error probability for both approaches, as follows:

**Theorem B.1.** *In the trajectory depicted in Fig. 9, the probability of the flat policy $\pi$ selecting an incorrect action is given as $\Omega(\pi) = \Phi\left(-\frac{\sqrt{2}}{\sigma\sqrt{T(T+2)}}\right)$ and the probability of the phasic policy structure $\pi_{subgoal} \circ \pi_{deisredgoal}$ selecting an incorrect action is bounded as $\Omega(\pi_{sg} \circ \pi_{dg}) \leq \Phi\left(-\frac{\sqrt{2}}{\sigma\sqrt{(T/i)^2 + 2(T/i)}}\right) + \Phi\left(-\frac{\sqrt{2}}{\sigma\sqrt{i(i+2)}}\right)$, where $\Phi$ denotes the cumulative distribution function of the standard normal distribution $\Phi(x) = \mathbb{P}[z \leq x] = -\frac{1}{\sqrt{2\pi}}\int_{-\infty}^{x} e^{-T^2/2} \mathrm{d}T$.*

**Proof.** Defining $z_1 := z_{1,T}$ and $z_2 := z_{1,T}$, the probability of the flat policy $\pi$ selecting an incorrect action can be computed as follows:

$$
\begin{aligned}
\Omega(\pi) &= \mathbb{P}\left[\hat{Q}(s+1, a, g) \leq \hat{Q}(s-1, a, g)\right] \\
&= \mathbb{P}\left[-T(1 + \sigma z_1) \leq -(T+2)(1 + \sigma z_2)\right] \\
&= \mathbb{P}\left[z_1 \sigma(T) - z_2 \sigma(T+2) \leq -2\right] \\
&= \mathbb{P}\left[z\sigma\sqrt{T(T+2)} \leq -\sqrt{2}\right] \\
&= \Phi\left(-\frac{\sqrt{2}}{\sigma\sqrt{T(T+2)}}\right),
\end{aligned}
\tag{31}
$$

where $z$ represent a standard Gaussian random variable. We leverage the property that the sum of two independent Gaussian random variables with standard deviations $\sigma_1$ and $\sigma_2$ results in a Gaussian distribution with a standard deviation of $\sqrt{\sigma_1^2 + \sigma_2^2}$.

Similar to the flat policy, the probability of the phasic policy selecting an incorrect action can be estimated using a union bound as follows:

$$\Omega(\pi_{subgoal} \circ \pi_{desiredgoal}) \leq \Omega(\pi_{subgoal}) + \Omega(\pi_{desiredgoal})$$

$$= \mathbb{P}\left[\hat{Q}(s+i,a,g) \leq \hat{Q}(s-i,a,g)\right] +$$

$$\mathbb{P}\left[\hat{Q}(s+1,a,s+i) \leq \hat{Q}(s-1,a,s+i)\right]$$

$$= \mathbb{P}\left[\hat{Q}(i,a,T) \leq \hat{Q}(-i,a,T)\right] + \mathbb{P}\left[\hat{Q}(1,a,i) \leq \hat{Q}(-1,a,i)\right]$$

$$= \Phi\left(-\frac{\sqrt{2}}{\sigma\sqrt{(T/i)^2 + 2(T/i)}}\right) + \Phi\left(-\frac{\sqrt{2}}{\sigma\sqrt{i(i+2)}}\right). \tag{32}$$

We observe that the error terms in the phasic goal-conditioned policy bound are consistently smaller than or equal to those in the flat policy. This implies that the accuracy of both the subgoal and desired goal policies surpasses that of the flat policy.

To evaluate the effectiveness of our phasic policy in selecting the correct actions, we conducted experiments on the gridworld environment, following the methodology outlined in Park et al. (2024). Specifically, we tested whether the policy learned by GCQS could reliably reach the desired goals. As illustrated in Fig. 11, under noisy Q-values, traditional flat policies do not always produce correct actions and may even generate erroneous actions, particularly in states far from the desired goal. In contrast, our GCQS approach demonstrates the ability to consistently generate correct policies that direct the agent towards the desired goals.

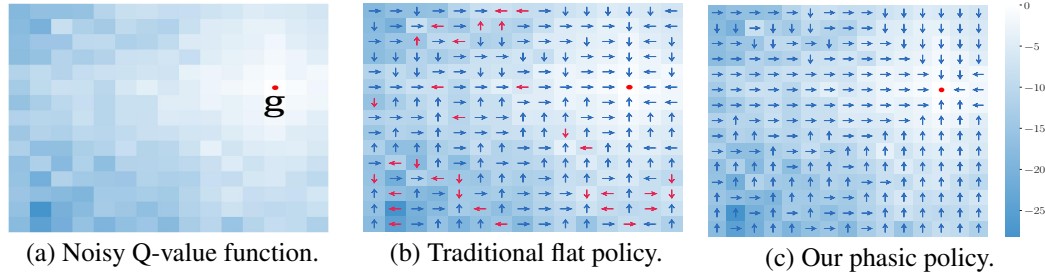

(a) Noisy Q-value function.  (b) Traditional flat policy.  (c) Our phasic policy.

Figure 11: The phasic policy structure in GCQS outperforms traditional flat policies in learning under noisy Q-values. $g$ is the desired goal. (a) Noisy Q-values are inherent in this gridworld environment. (b) The traditional flat policy is prone to producing incorrect actions ($\rightarrow$), especially in states that are far from the desired goal. (c) The phasic policy is still able to produce correct actions ($\rightarrow$), thanks to the subgoal mechanism.

## C EXPERIMENTAL DETAILS

In this section, we provide experimental details omitted in Section 6 of the main paper. These include (1) technical and architecture details for all methods, (2) experimental evaluating setup, (3) hyperparameters for all methods.

### C.1 ALGORITHM AND ARCHITECTURE

We employ the off-policy actor-critic algorithm with HER (Andrychowicz et al., 2017) as our foundational goal-conditioned RL framework. This sequential comparison experiment allows us to directly assess the relative performance and effectiveness of each approach under identical conditions. Additionally, temporal difference (TD) learning is utilized for value function estimation, and soft updates are applied to network parameters. Our implementation adheres to the optimal parameter settings as outlined in Plappert et al. (2018). The hyperparameters for all baseline methods remain consistent. For GCQS, the policy objective parameter $\beta$ is set to 0.2. For further details, refer to

Appendix D.2. Our implementation of baselines and GCQS draw knowledge from and references the following four code repositories:

- DDPG, DDPG+HER, MHER, GCSL, WGCSL: `https://github.com/Cranial-XIX/metric-residual-network`;
- Actionable Models, GoFar: `https://github.com/JasonMa2016/GoFAR`;
- DWSL: `https://github.com/jhejna/dwsl/`;
- RIS: `https://github.com/elliotchanesane31/RIS`;
- BEAG, PIG, HIGL, CQM: `https://github.com/ml-postech/BEAG`;

Notably, although GoFar and DWSL are offline goal-conditioned methods, Yang et al. (2023) and (Hejna et al., 2023) indicate that they are both derived from Advantage-Weighted Regression (AWR) (Peng et al., 2019). Therefore, we re-implemented them, and they remain effective in the online setting.

## C.2 EVALUATION SETUP

For each baseline and task, we conducted evaluations using random five seeds (e.g, $\{100, 200, 300, 400, 500\}$). The policy was trained for 1000 episodes per epoch. Upon completing each training epoch, the policy's performance was measured by calculating the mean success rate from 100 independent rollouts, each using randomly selected desired goals. These success rates were averaged across five seeds and plotted over the learning epochs, with the standard deviation illustrated as a shaded region on the performance figure.

## C.3 EXPERIMENTAL HYPERPARAMETERS

We consistently utilize the Adam optimizer (Kingma, 2014) across all experimental setups. For each state, goals are uniformly relabeled by sampling from all future states within its trajectory. In environments applying discount factors, we set $\gamma = 0.98$ for all goal-conditioned tasks. Each algorithm follows a predetermined set of hyperparameters specifically designed for goal-conditioned environments. PIG, DWSL, GoFar, WGCSL, GCSL, MHER, and DDPG have been previously calibrated for our task set, and we have adopted the parameter values as reported in prior research. Our implementation of PIG shares the same network architecture as DDPG, thus utilizing DDPG's hyperparameter values. Detailed hyperparameter configurations used in this study are provided in Table 1, which have been identified through the aforementioned parameter search process.

## C.4 ENVIRONMENT DETAILS

In this section, we describe the tasks in our experiments in Section 6. All goal-conditioned tasks are derived from OpenAI Gym (Brockman, 2016).

**Fetch Tasks**   The Fetch tasks (i.e, FetchReach, FetchPush, FetchSlide, FetchPick), involve controlling a 7-DoF robotic arm to complete various goal-directed actions such as reaching, pushing, sliding, or picking up an object and moving it to a target location. These environments share common characteristics, including a multidimensional state space that represents the arm's position and velocities, and a 4-dimensional action space for movement and gripper control. The tasks are goal-conditioned, with the reward function defined by whether the arm or object reaches the desired goal within an allowable margin of error. The main variation between tasks lies in the specific goal (i.e, reaching, pushing, or picking) and the interactions with the object, such as sliding it beyond the robot's direct reach or placing it at a target on the table or in the air. The allowable error in Fetch tasks is $\mu = 0.05$. The reward function is defined as:

$$r(s, a, g_{XYZ}) = 1(\|\phi(s) - g_{XYZ}\|_2^2 \leq \mu).$$

**Hand Tasks**   The Hand tasks (i.e, HandReach, BlockRotateZ, BlockRotateXYZ, BlockRotatePar­allel) focus on controlling a Shadow Dexterous Hand to manipulate objects in high-dimensional tasks, requiring precise control over 20 independent joints. Each task features complex observations,

Table 1: Hyperparameters for Baselines.

| Actor and critic networks | Value |
|---|---|
| Learning rate | 1e-3 |
| Buffer size | $10^6$ transitions |
| Polyak-averaging coefficient | 0.95 |
| Action L2 norm coefficient | 1.0 |
| Observation clipping | [-200,200] |
| warmup steps | 5000 |
| Batch size | 256 |
| Rollouts per MPI worker | 2 |
| Number of MPI workers | 16 |
| Cycles per epoch | 50 |
| Batches per cycle | 40 |
| Test rollouts per epoch | 10 |
| Probability of random actions | 0.3 |
| Scale of additive Gaussian noise | 0.2 |
| Probability of HER experience replay | 0.8 |
| Normalized clipping | [-5, 5] |
| $\beta$ | 0.2 |

including joint positions, velocities, and object state information (position, rotation, and velocities). The reward structure is sparse and binary, with goals achieved when the object reaches a specified position or rotation within a defined tolerance. These tasks, which vary in manipulation complexity (e.g., specific axis rotations), present a challenging testbed for advanced goal-conditioned reinforcement learning algorithms in high-dimensional control settings. The reward function is the same as Fetch tasks and the allowable threshold ($\mu = 0.01$).

**AntMaze Tasks** A quadruped ant robot is trained to reach a random goal from a random location and tested under the most difficult setting for each maze. The states of ant is 30-dimension, including positions and velocities. An ant should reach the target point within 500 steps for U-shaped mazes, and 1000 steps for S-, $\omega$-, and $\Pi$-shaped mazes. The reward function is the same as Fetch tasks and the allowable threshold ($\mu = 0.1$).

# D  ADDITIONAL RESULTS

This section evaluates the resilience of GCQS across several factors, including the number of subgoals, the hyperparameter $\beta$, robustness to environmental stochasticity, and the relabeling ratio. Due to space limitations, not all of these variations were discussed in the main body of this study. These details are provided below.

## D.1  THE IMPACT NUMBER OF SUBGOALS

In our approach, subgoals play a pivotal role, and thus, apart from their selection, investigating the optimal quantity of subgoals is imperative. We systematically vary the proportion of subgoals selected from $\tau^{g'}$ relative to the total trajectory goals, and benchmark these against competitive algorithms such as WGCSL, DDPG+HER, GoFar, and DWSL. We evaluate algorithmic performance across four different subgoal proportions $\{20\%, 50\%, 90\%, 100\%\}$. Analysis presented in Fig. 12 demonstrates

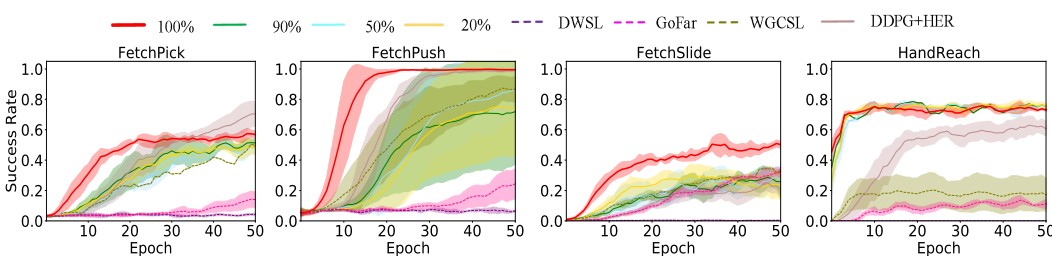

Figure 12: Subgoal number ablation studies in some goal-conditioned tasks.

that GCQS consistently surpasses the performance of the aforementioned algorithms, regardless of the percentage of subgoals employed. This finding highlights the robustness of our method in response to variations in the quantity of subgoals utilized.

## D.2 THE IMPACT OF HYPERPARAMETER $\beta$

Since the addition of KL regularization term in the policy improvement stage of our method (as shown in Eq. (15)), this section explores the influence of the balancing parameter $\beta$. We evaluate $\beta$ values from the set $\{0.2, 0.5, 1.0, 3.0\}$ and compare the results against competitive HER-based algorithms such as WGCSL and DDPG+HER, as shown in Fig. 13. The findings in Fig. 13 reveal that GCQS consistently delivers superior performance over the other algorithms, regardless of the $\beta$ parameter variation. This demonstrates that our method maintains robustness and is not significantly affected by changes in the $\beta$ parameter.

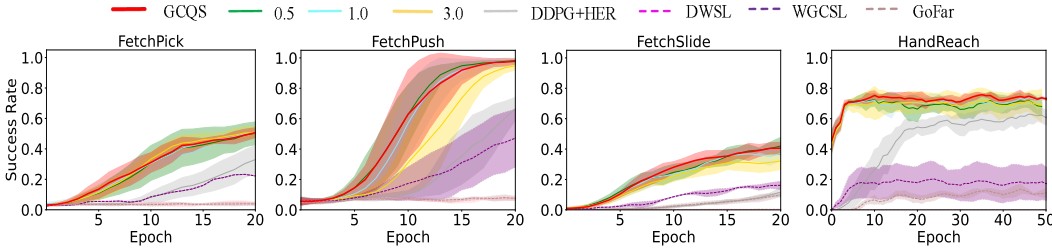

Figure 13: Hyperparameter $\beta$ ablation studies in such goal-conditioned tasks.

## D.3 ERROR BARS OF MEAN PERFORMANCE

To further assess the effectiveness and robustness of the algorithm, we present error bar plots for each task based on the mean ± standard deviation (SD) of results across five seeds for each algorithm. As shown in Fig. 14, the GCQS algorithm demonstrates a significant advantage across all goal-conditioned tasks. Its mean success rate approaches 100% on simpler tasks (e.g., FetchReach and BlockRotateZ), and it substantially outperforms other algorithms on moderately challenging tasks (e.g., FetchPush and HandReach), with shorter error bars indicating greater result stability and robustness. However, in more difficult tasks (e.g., BlockRotateXYZ and BlockRotateParallel), the performance of GCQS declines, as evidenced by lower success rates and longer error bars, suggesting performance fluctuations. Overall, GCQS exhibits strong learning capabilities in complex goal spaces but still has room for improvement, particularly in handling extreme tasks such as high-dimensional rotations and parallel rotations.

## D.4 ROBUST TO ENVIRONMENTAL STOCHASTICITY

To test whether our GCQS is robust to random environmental factors, we follow GoFar's settings. Specifically, we examine a modified FetchPush environment characterized by the introduction of Gaussian noise with a zero mean before action execution. This modification generates various

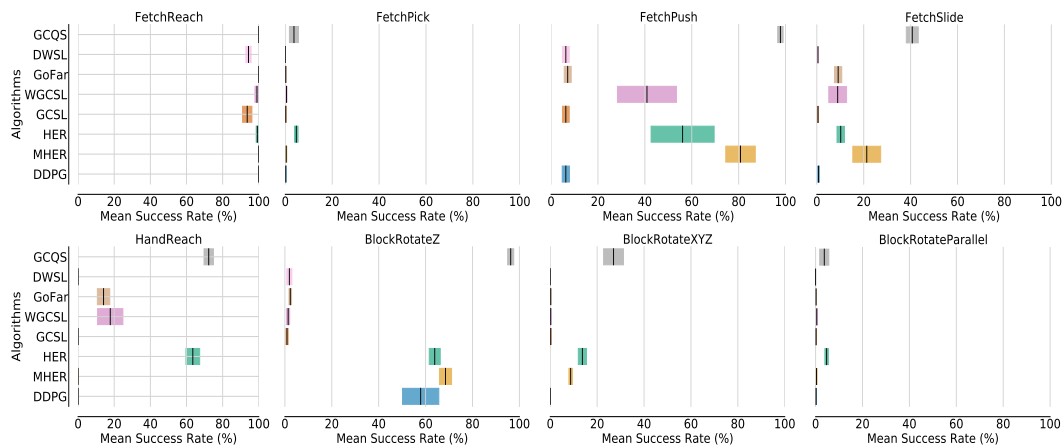

Figure 14: The error bars for each goal-conditioned task presented in Fig. 5. Error bars represent the standard error of the mean (SEM) for each algorithm's average performance across multiple seeds in each task.

environmental conditions with standard deviations of $\{0.2, 0.5, 1.0, 1.5\}$, allowing us to analyze the robustness and performance of the proposed method under differing levels of stochasticity.

As we see in Fig. 15, GCQS is the most robust to stochasticity in the FetchPush environment, also outperforming baseline algorithms in terms of mean success rate under various noise levels. WGCSL exhibits minimal sensitivity to variations in all noise levels, whereas DDPG+HER is moderately sensitive. At a noise level of 0.5, the performance gap continues to widen, with GoFar exhibiting a significant collapse, underscoring its heightened sensitivity to noise. Despite DWSL's insensitivity to noise, its overall performance remains suboptimal. Overall, the phsic structural policy optimization in GCQS indeed confers greater robustness to environmental stochasticity.

We suggest that the assumption of deterministic dynamics embedded in self-supervised learning methods, such as WGCSL, GoFar, and DWSL, may lead to overly optimistic performance assessments in stochastic environments. In contrast, reinforcement learning methods have the ability to effectively adapt to these stochastic changes.

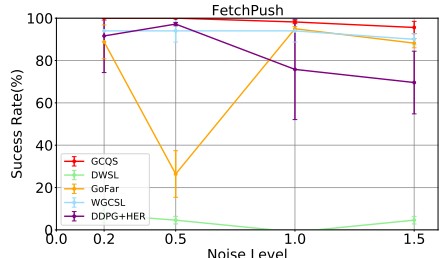

Figure 15: Mean success rate (%) for FetchPush task under environment stochasticity.

### D.5 SAMPLE EFFICIENCY

To assess the sample efficiency of baseline methods in comparison to GCQS, we examined the number of training samples (i.e., $\langle s, a, g', g \rangle$ tuples) necessary to obtain a particular mean success rate. This comparative analysis is depicted in Fig. 16.

From the FetchPush task, depicted on the left side of Fig. 16, we observe that to attain the 0.45 mean success rate, the competitive baseline DDPG+HER requires over 6000 training samples, whereas GCQS only needs approximately 4000 samples. This indicates that GCQS is 1.5 times more sample efficient than DDPG+HER.

In another task, BlockRotateZ, GCQS uses the fewest number of samples to attain the same 0.5 mean success

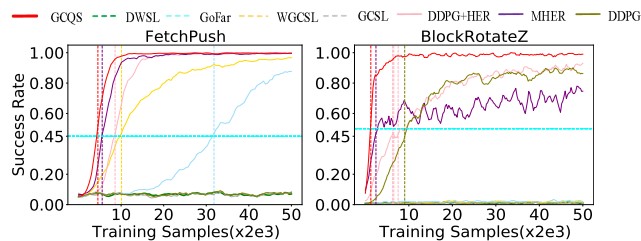

Figure 16: Number of training samples needed with respect to mean success rate for Fetchpush and HandManipulate-BlockRotateZ tasks (the lower the better).

rate. These findings demonstrate that
GCQS significantly enhances sample efficiency compared to other baseline methods, underscoring its effectiveness in improving learning performance with fewer training samples.

## D.6    RELABELING RATIO

Given our approach to learning in goal-conditioned RL settings, which assumes data annotated with relabeled goals, this study examines the influence of explicit goal labels on performance. We conducted experiments across four distinct relabeling ratios (i.e, $0.2, 0.5, 0.8, 1.0$) in various environments to evaluate algorithmic efficacy. As illustrated in Fig. 17, GCQS exhibits substantial resilience to variations in the relabeling ratio. Furthermore, GCQS consistently surpasses competing algorithms such as WGCSL and DDPG+HER across different labeling ratios.

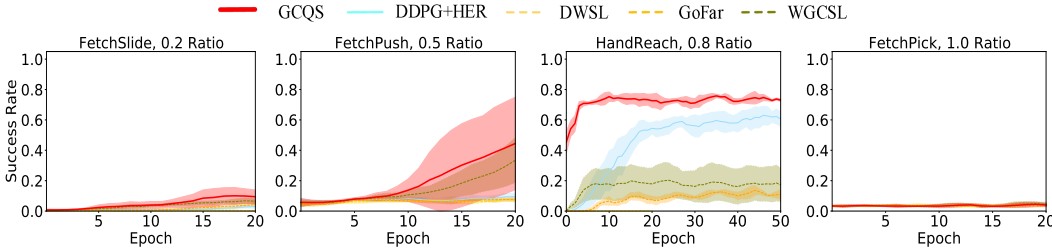

Figure 17: Relabel ratio ablation studies in some goal-conditioned tasks.

