# OpenReview forum: "Goal-conditioned Reinforcement Learning with Subgoals Generated from Relabeling"
_ICLR.cc/2025/Conference — Submitted to ICLR 2025_

### Official Review · Reviewer_cURz · 2024-10-19

**Soundness:** 2
**Presentation:** 2
**Contribution:** 2
**Rating:** 5
**Confidence:** 3

**Summary:**

This paper studies a tendency in existing GCRL methods with goal relabelling that prioritises optimisation toward closer achieved goals. To mitigate this bias, the authors propose an actor-critic objective without goal relabelling, incorporating a KL-divergence constraint towards a novel prior policy. This prior policy ensures that the learned policy behaves consistently towards both the final desired goal and intermediate subgoals. In the experiments, the authors demonstrate that the proposed method achieves strong performance with fewer environment interactions, although the experimental design remains contentious.

**Strengths:**

1. The motivation behind the proposed method is insightful and articulated clearly, providing a strong rationale for the approach.
2. The method developed from this motivation is based on a promising and well-supported assumption.

**Weaknesses:**

The main concern with this paper lies in the experimental design:
1. Section 6.1: It lacks comparison with suitable baselines (see Question 5).
2. Section 6.2: This section appears redundant, as GCQS's advantage in sample efficiency is already demonstrated in Figure 5 and Section 6.1.

Some minor issues are noted in the questions section.

**Questions:**

1. The definition of Q-BC should be clarified upon its first appearance (line 80).
2. There is a typo in line 312-313 - 'policy objective that reaching achieved goals'.
2. Section 2 discusses GCWSL methods like GoFar (Ma et al., 2022) and WGCSL (Yang et al., 2022) without mentioning their offline nature. Yang et al. prove the theoretical guarantees in the offline goal-conditioned setting—do these guarantees still hold in the online setting?
3. In Section 5.2, the relabelled (achieved) goal $g'$ is redefined as subgoals $s_g$, but both $g'$ and $s_g$ are used in the text, which causes some confusion. Would it be clearer to consistently use the notation $g'$ throughout the paper?
5. The comparison in Section 6.1 appears unfair, as some baselines, like Actionable Models, WGCSL, and GoFar, are designed for offline goal-conditioned RL and may not perform as well in the online setting.

**References**

Ma, J. Y., Yan, J., Jayaraman, D., \& Bastani, O. (2022). Offline Goal-Conditioned Reinforcement Learning via f-Advantage Regression. NeurIPS.

Yang, R., Lu, Y., Li, W., Sun, H., Fang, M., Du, Y., … Zhang, C. (2022). Rethinking Goal-conditioned Supervised Learning and Its Connection to Offline RL. ICLR.

---

> ### Author Response · Authors · 2024-11-20
> **Author Response**
>
> Thank you for the detailed review and for the suggestions for improving the work. Below, I have carefully addressed your comments and concerns.
>
> **1. Section 6.1: It lacks comparison with suitable baselines (see Question 5).**
>
> Thank you for raising this issue. We will address it in Responses 5.
>
> **2. Section 6.2: This section appears redundant, as GCQS's advantage in sample efficiency is already demonstrated in Figure 5 and Section 6.1.**
>
> Thank you for your suggestion. We have moved Section 6.2 to the appendix and revised it to include a comparison with the state-of-the-art algorithm, PIG [1].
>
> **3. The definition of Q-BC should be clarified upon its first appearance (line 80).**
>
> Thanks for the suggestions here. We have done this.
>
> **4. There is a typo in line 312-313 - 'policy objective that reaching achieved goals'.**
>
> Thanks for the suggestion – we have fixed this.
>
> **5. Section 2 discusses GCWSL methods like GoFar (Ma et al., 2022) and WGCSL (Yang et al., 2022) without mentioning their offline nature. Yang et al. prove the theoretical guarantees in the offline goal-conditioned setting—do these guarantees still hold in the online setting?**
>
> Firstly, the WGCSL [2] paper indicates (i.e., in line 14 of the abstract and the results in Figure 15) that WGCSL is applicable to both online and offline settings. Secondly, [3] indicates that WGSL and GoFar belong to the family of Advantage-weighted Regression (AWR). Finally, the work in [4] also highlights the relationship between DWSL and AWR. Given that AWR [5] is effective in online settings, we believe these baselines are also applicable in the online setting. Additionally, ActionModels [6] combines goal-chaining with conservative Q-Learning, which, as noted, is not suitable for the offline setting. Consequently, we have removed this baseline in the revised version. In the revised version, we have included a comparison with the subgoal-based state-of-the-art method, PIG [1]. The results indicate that our GCQS method can achieve performance comparable to the state-of-the-art method PIG [1], even without the additional algorithmic design for planning or subgoal discovery.
>
> **6. In Section 5.2, the relabelled (achieved) goal $g'$ is redefined as subgoals $s_g$, but both $g'$ and $s_g$ are used in the text, which causes some confusion. Would it be clearer to consistently use the notation $g'$ throughout the paper?**
>
> Thank you for your suggestions regarding the expression in the paper. $g'$ represents the achieved goals, which are relabeled targets, while $s_g$ denotes subgoals sampled from $g'$. Although they may share the same physical meaning in practice, their conceptual representations differ. Specifically, $g'$ refers to relabeled data, whereas $s_g$ can be understood as goals sampled from the relabeled data, serving as essential waypoints toward reaching the desired goal. The distinction made in Section 5.2 emphasizes this difference to provide clarity.
>
> **7. The comparison in Section 6.1 appears unfair, as some baselines, like Actionable Models, WGCSL, and GoFar, are designed for offline goal-conditioned RL and may not perform as well in the online setting.**
>
> Thank you for raising this issue. We have addressed it in Response 5.  Additionally, we have included a comparative experiment with online-related subgoal sota algorithm in Section 6.2.
>
> [1] Kim, Junsu, et al. "Imitating graph-based planning with goal-conditioned policies." arXiv preprint arXiv:2303.11166 (2023).
>
> [2] Yang, R., Lu, Y., Li, W., Sun, H., Fang, M., Du, Y., … Zhang, C. (2022). Rethinking Goal-conditioned Supervised Learning and Its Connection to Offline RL. ICLR.
>
> [3] Yang, Wenyan, et al. "Swapped goal-conditioned offline reinforcement learning." arXiv preprint arXiv:2302.08865 (2023).
>
> [4] Hejna, Joey, Jensen Gao, and Dorsa Sadigh. "Distance weighted supervised learning for offline interaction data." International Conference on Machine Learning. PMLR, 2023.
>
> [5] Peng, Xue Bin, et al. "Advantage-weighted regression: Simple and scalable off-policy reinforcement learning." arXiv preprint arXiv:1910.00177 (2019).
>
> [6] Chebotar, Yevgen, et al. "Actionable models: Unsupervised offline reinforcement learning of robotic skills." arXiv preprint arXiv:2104.07749 (2021).

---

> > ### Comment · Reviewer_cURz · 2024-11-24
> >
> > Thank you for your response. However, I believe there is still room for improvement to further strengthen this paper. For instance, adding more baselines to the long-horizon Antmaze tasks could enhance the evaluation, and providing additional evidence to support the argument in Lines 106–107 would be beneficial. While this does not require an ablation study, it would be ideal to include visualizations that clearly illustrate what happens. I maintain my current rating.

---

> > > ### Author Response · Authors · 2024-12-03
> > >
> > > We appreciate the time and effort you have dedicated to our paper. After brief discussions, concerns have been continuously addressed, and the paper has been progressively refined. If the reviewer feels that the concerns have been largely resolved, we hope for a rating increase as a happy ending to this series of discussions.
> > >
> > > Regardless, we wish the reviewer all the best in both research and life!

---

> ### Author Response · Authors · 2024-12-01
> **Author Response**
>
> Thank you for your valuable questions and suggestions. In response, we have incorporated several other state-of-the-art subgoal-based baseline algorithms, such as BEAG [1], DHRL [2], and HIGL [3], into the complex long-horizon Antmaze tasks. In Section 6.2 of the revised version [4],  results in long-horizon Antmaze indicate that GCQS demonstrates performance that is slightly inferior to or comparable with PIG, while outperforming both HIGL and DHRL. Additionally, GCQS achieves twice the training speed of PIG and three times that of BEAG, with significantly reduced memory costs.
>
> Additionally, we have provided a visual demonstration for the claims made in lines 106-107. In Figure 11 of the revised version, we highlight the advantages of the phasic policy structure in the Gridworld environment. Under noisy Q-values, traditional flat policies do not always yield correct actions and may even generate erroneous ones, particularly in states far from the desired goal. In contrast, our GCQS approach consistently produces correct policies that guide the agent towards the desired goals.
>
> [1]Yoon, Youngsik, et al. "Breadth-First Exploration on Adaptive Grid for Reinforcement Learning." ICML 2024.
>
> [2]Lee, Seungjae, et al. "DHRL: a graph-based approach for long-horizon and sparse hierarchical reinforcement learning." NIPS 2022.
>
> [3]Kim, Junsu, Younggyo Seo, and Jinwoo Shin. "Landmark-guided subgoal generation in hierarchical reinforcement learning." NIPS 2021.
>
> [4]  https://pdfhost.io/v/~QVHMgEkR_ICLR2025_GCQS_Rebuttal.
>
> [5]Kim, Junsu, et al. "Imitating graph-based planning with goal-conditioned policies." ICLR 2023.

---

### Official Review · Reviewer_kNZr · 2024-11-02

**Soundness:** 2
**Presentation:** 2
**Contribution:** 2
**Rating:** 5
**Confidence:** 4

**Summary:**

This paper works on goal-conditioned reinforcement learning tasks and introduces the idea of leveraging subgoals in hindsight relabeling to improve learning efficiency. The proposed method is mainly heuristic, and it is justified with theory on the performance. Empirical results are provided to support the method.

**Strengths:**

The authors appropriately used illustrative figures to explain some concepts in their paper. Motivating examples are provided to demonstrate the problem of previous approaches. The experiments are conducted on multiple tasks and results are compared with multiple baselines.

**Weaknesses:**

There are several main weaknesses in the current paper:

1. on the high level, the necessity of the proposed method is not clear. I understand the key insight the authors aimed to convey using Figure 2. However, from my perspective, isn't this a *property* rather than a *problem* of hindsight relabeling? e.g. when the number of steps required to achieve different goals is uniformly sampled from 1-50, then during the learning process, hindsight relabeling using trajectories having a length of 50 will generate a uniformly distributed length from 1-50, and as the learning proceeds, the policy will learn to achieve closer goals, and averaged trajectory length will be smaller than 50 --- this will lead to a compounding effect in increasingly having more shorter paths.


2. to solve the above challenge, why is the proposed method "necessary"? For instance, why can not one just to re-sample or down-sample on some of the state-goals? In the current write-up, it is not very clear why the multiple designing factors introduced are well-motivated or necessary.


3. according to the theory, using a smaller $\eta$ will lead to a tighter bound, does this mean that we need the $\eta$ to be as close to 0 as possible?

4. In Figure 5, the averaging/smoothing method for different figures seems different, and some results lack error bars. The FetchSlide performance is noticeably lower than what has been achieved in the literature.
From my perspective, it is not necessary to demonstrate a method is always better than baselines in all experimental cases. What is more important is to clearly show what are the realistic motivations for introducing the techniques/designing components of the paper, and then use experiments to highlight --- the settings where those challenges exist, the method shines and on the other hand, when those challenges (controllably) alleviated, the performances of different methods would converge.


Minor:

5. On presentation. In Figure 1, it would be great if the authors could give some concrete examples and make sure the notations are self-consistent. I acknowledge the authors' effort in explaining their method using Figure 1 and it could be a great idea if the clarity can be further enhanced. For instance, what is "QBC" objective in this figure, what does it mean by "subgoals come from achieved goals", and why the policy can take either s and g or s and s_g as its inputs?

**Questions:**

please refer to weaknesses.

---

> ### Author Response · Authors · 2024-11-20
> **Author Response**
>
> We thank the reviewer for the detailed review. Below, we address the raised concerns and questions.
>
> **1.on the high level, the necessity of the proposed method is not clear. I understand the key insight the authors aimed to convey using Figure 2. However, from my perspective, isn't this a property rather than a problem of hindsight relabeling? e.g. when the number of steps required to achieve different goals is uniformly sampled from 1-50, then during the learning process, hindsight relabeling using trajectories having a length of 50 will generate a uniformly distributed length from 1-50, and as the learning proceeds, the policy will learn to achieve closer goals, and averaged trajectory length will be smaller than 50 --- this will lead to a compounding effect in increasingly having more shorter paths.**
>
> Existing HER-based methods tend to prioritize shorter segments of trajectories during the update process, meaning that updates often utilize short-length goals from the same trajectory, while longer segments remain unused and this valuable information is completely overlooked. **Our perspective emphasizes leveraging longer trajectory information within the same hindsight framework.** To achieve this, we have developed a novel algorithm called GCQS, which can be viewed as an extended version of GCAC. Specifically, GCQS leverages the achieved goals trajectory as the subgoal distribution. As illustrated in **Figure 6**, we analyzed the trajectory information utilized by GCQS during the update process. The results indicate that GCQS effectively incorporates trajectory data of varying lengths, showcasing strong performance across different scenarios.
>
> **2. to solve the above challenge, why is the proposed method "necessary"? For instance, why can not one just to re-sample or down-sample on some of the state-goals? In the current write-up, it is not very clear why the multiple designing factors introduced are well-motivated or necessary.**
>
> Re-sampling or down-sampling does not necessarily enable the use of longer trajectories for updates. In contrast, our GCQS framework is designed to leverage longer trajectories during the update process effectively.
>
> **3. according to the theory, using a smaller $\eta$ will lead to a tighter bound, does this mean that we need the  \eta to be as close to 0 as possible?.**
>
> In Equation (15), the KL divergence approaches zero only when $\beta$ approaches positive infinity. Otherwise, it is unlikely to converge to zero. In our work, $\beta$ is set to 0.2, which means the KL divergence does not approach zero, and consequently,
> $\eta$ is also not close to zero. As demonstrated in the ablation study (Figure 12), increasing $\beta$ does not necessarily lead to better performance, indicating that setting $\beta$ to positive infinity would be suboptimal.
>
> **4. In Figure 5, the averaging/smoothing method for different figures seems different, and some results lack error bars. The FetchSlide performance is noticeably lower than what has been achieved in the literature. From my perspective, it is not necessary to demonstrate a method is always better than baselines in all experimental cases. What is more important is to clearly show what are the realistic motivations for introducing the techniques/designing components of the paper, and then use experiments to highlight --- the settings where those challenges exist, the method shines and on the other hand, when those challenges (controllably) alleviated, the performances of different methods would converge.**
>
> Thank you for your suggestion. In the revised version of the paper, we have included the error bar plots for Figure 5 in Section D.4 of the Appendix.
>
> Our implementation is based on the official code from [1], and similarly to [1], the FetchSlide task exhibits a relatively low mean success rate, which may be due to suboptimal hyperparameter tuning in this environment.
>
> **5. On presentation. In Figure 1, it would be great if the authors could give some concrete examples and make sure the notations are self-consistent. I acknowledge the authors' effort in explaining their method using Figure 1 and it could be a great idea if the clarity can be further enhanced. For instance, what is "QBC" objective in this figure, what does it mean by "subgoals come from achieved goals", and why the policy can take either s and g or s and s_g as its inputs?**
>
> Thank you for your suggestion. In the revised version, we have updated the phrasing for clarity. Additionally, the Q-BC objective is explained in Section 5.1. The phrase "subgoals come from achieved goals" means that subgoals are sampled from achieved goals (i.e., relabeled goals). Specifically, (s,g) serve as the inputs to the policy $\pi$, while (s,s_g) are the inputs to the prior policy $\pi^{piror}$.
>
> [1] Liu, Bo, et al. "Metric Residual Network for Sample Efficient Goal-Conditioned Reinforcement Learning." Proceedings of the AAAI Conference on Artificial Intelligence. Vol. 37. No. 7. 2023.

---

> > ### Comment · Reviewer_kNZr · 2024-11-25
> > **Thank you for the response.**
> >
> > I appreciate the authors' response. For Q4 and Q5 the response seems okay to me (though I'm not a fan of using sub-optimal implementations and then attributing the performance gap to previous work. The literature should be criticized rather than followed.)
> >
> > However, Q1 and Q3 are not directly answered. For Q2, sorry but I'm still not clear why is it *necessary*, well-controlled ablation studies and empirical comparisons on the necessity of those designing factors could be helpful.

---

> > > ### Author Response · Authors · 2024-12-03
> > >
> > > As the author-reviewer discussion phase is coming to an end, we are eagerly awaiting your feedback on whether your previous concerns have been addressed and if there are any further questions. If all your concerns have been resolved, we hope that you can consider raising the rating accordingly.
> > >
> > > Regardless, we wish you all the best!

---

> ### Author Response · Authors · 2024-12-01
> **Author Response**
>
> We appreciate the reviewer’s comments and feedback.
>
> For Q1 and Q2:
>
> As discussed in Section 4, we concluded that baseline algorithms often rely on short trajectories for updates (due to uniform sampling of achieved goals). Therefore, addressing this issue could potentially lead to better performance. Following the approach suggested by reviewer JyB4, we implemented simple weighted sampling, or upsampling (where longer trajectories receive higher weights), which favors longer trajectories. The results are presented in Appendix D.1 of the revised version [1]. Our findings show that while simple weighted sampling (or upsampling) does improve the performance of the baseline algorithms, it still does not outperform GCQS. This highlights the importance and necessity of the subgoal mechanism in GCQS, which enables the learning of more optimal actions.
>
> For Q3:
>
> No, in Equation 15, when the KL term equals 0, it leads to a trivial solution, meaning that the policy no longer updates. Instead, the policy should be updated under a smaller value of
> \eta, which allows for gradual policy updates.
>
> [1] https://pdfhost.io/v/~QVHMgEkR_ICLR2025_GCQS_Rebuttal

---

### Official Review · Reviewer_kpwt · 2024-11-03

**Soundness:** 3
**Presentation:** 3
**Contribution:** 2
**Rating:** 5
**Confidence:** 3

**Summary:**

This paper presents GCQS, which utilizes achieved goals as subgoals and iteratively updates by considering behavior cloning schemes for prior policy and KL-constraint for policy.

**Strengths:**

This paper is generally well-written, easy to follow, and provides promising results on some benchmark problems.
The paper also highlights neglected aspects, such as the issue of short trajectory usage in subgoal generation, and presents theoretical support for the proposed objective.

**Weaknesses:**

The lack of comparison with other benchmark problems, such as complex AntMaze tasks presented in [1, 2], where the original HER showed significantly degraded performance compared to similar methods such as PIG, raises questions about whether omitting high-level planning for subgoal generation is valid for other complex and long-horizon GCRL tasks.

See questions for others.

[1] Kim, Junsu, et al. "Imitating graph-based planning with goal-conditioned policies." arXiv preprint arXiv:2303.11166 (2023).

[2] Kim, Junsu, Younggyo Seo, and Jinwoo Shin. "Landmark-guided subgoal generation in hierarchical reinforcement learning." Advances in neural information processing systems 34 (2021): 28336-28349.

**Questions:**

(1) This is the question for clarification. What is the key difference between the proposed methods compared to existing methods [1,2] which update subgoals throughout training? Specifically, how does relabelling achieved goals as subgoals differ from landmark or subgoal updates in PIG and HIGL?

(2) This method discards the necessity of high-level planning for subgoal generation but I'm curious whether GCQS works well in long-horizon tasks such as AntMaze [1,2,3] compared to methods utilizing high-level planning, such as HIGL and PIG.

(3) It is unclear how GCQS addresses the issue of short trajectory updates since HER already conducts a similar relabeling scheme. Which component of GCQS handles this?

(4) How important is the SAC component in Eq.12 for GCQS? This omitted ablation leaves the question of the importance of the BC term.

(5) Presenting more details on the derivation of Eq. 25 from Eq. 24 would be helpful for readers.

[1] Kim, Junsu, et al. "Imitating graph-based planning with goal-conditioned policies." arXiv preprint arXiv:2303.11166 (2023).

[2] Kim, Junsu, Younggyo Seo, and Jinwoo Shin. "Landmark-guided subgoal generation in hierarchical reinforcement learning." Advances in neural information processing systems 34 (2021): 28336-28349.

[3] Lee, Seungjae, et al. "Cqm: Curriculum reinforcement learning with a quantized world model." Advances in Neural Information Processing Systems 36 (2023): 78824-78845.

---

> ### Author Response · Authors · 2024-11-20
> **Author Response (1/2)**
>
> We sincerely thank the reviewer for their thorough review and valuable feedback on our work.
>
> **1.The lack of comparison with other benchmark problems, such as complex AntMaze tasks presented in [1, 2], where the original HER showed significantly degraded performance compared to similar methods such as PIG, raises questions about whether omitting high-level planning for subgoal generation is valid for other complex and long-horizon GCRL tasks.**
>
> Thank you for your suggestion. In the revised version, we have included a comparison with the subgoal-based state-of-the-art method, PIG [1], to enhance the analysis.
>
> **2. This is the question for clarification. What is the key difference between the proposed methods compared to existing methods [1,2] which update subgoals throughout training? Specifically, how does relabelling achieved goals as subgoals differ from landmark or subgoal updates in PIG and HIGL?**
>
> Our GCQS method does not involve designing additional strategies for planning or discovering subgoals. Instead, it directly selects subgoals from the relabeled achieved goals, aiming to leverage more information from the same trajectory (particularly in long trajectories).
>
> The key difference between the proposed GCQS method and existing methods such as PIG [1] and HIGL [2] lies in the approach and mechanism used for subgoal selection and integration:
>
> (1). Subgoal Discovery and Integration:
>
>    GCQS: Utilizes achieved goals directly from the agent's own trajectories as subgoals, leveraging hindsight relabeling. The method reuses these achieved goals within the same trajectory to enhance policy learning. This approach does not involve any additional discovery or planning mechanisms for subgoal identification. Instead, GCQS systematically redefines naturally occurring achieved goals from replayed trajectories, thus simplifying the integration of subgoals into the training process.
>
>    PIG and HIGL: Both incorporate explicit subgoal discovery and planning algorithms.
>
>    PIG: Uses a graph-based planning algorithm to identify subgoals and updates these subgoals dynamically throughout training. This method involves complex planning to guide the agent's behavior by imitating subgoal-conditioned policies and includes a strategy for subgoal skipping to optimize paths.
>
>    HIGL: Focuses on identifying landmarks that serve as promising exploration targets. It selects subgoals based on criteria like novelty and coverage to promote more efficient exploration and policy updates. The selected subgoals or landmarks are used in a hierarchical structure that requires additional computation for their discovery and validation.
>
> (2). Relabeling vs. Dynamic Updates:
>
>    GCQS's Relabeling: In GCQS, the subgoals are achieved goals that are relabeled post-experience. The relabeling mechanism converts observed outcomes into intermediate objectives without modifying or dynamically updating them based on external algorithms. This relabeling is derived directly from the agent's own trajectory, making it straightforward and computationally less intensive.
>
>    PIG and HIGL's Dynamic Updates: These methods involve active updates and the strategic placement of subgoals or landmarks during training. PIG dynamically plans and modifies subgoals during policy updates, whereas HIGL uses a novelty-based mechanism to adjust the subgoal selection process. These processes aim to guide exploration and learning more explicitly and involve more complex planning steps.
>
> (3). Complexity and Computational Cost:
>
>    GCQS maintains simplicity by leveraging the natural data produced by the agent's experience for subgoal selection, avoiding additional planning or search mechanisms.
>
>    PIG and HIGL incur a higher computational cost due to their need for subgoal discovery mechanisms, such as graph-based planners (PIG) or coverage/novelty-based sampling (HIGL).
>
> In summary, GCQS differentiates itself by using achieved goals as subgoals directly through hindsight relabeling, which simplifies the training process and reduces computational overhead. In contrast, PIG and HIGL involve more sophisticated subgoal or landmark discovery and updating mechanisms that require additional planning and computational resources to guide training effectively.

---

> ### Author Response · Authors · 2024-11-20
> **Author Response (2/2)**
>
> **3. This method discards the necessity of high-level planning for subgoal generation but I'm curious whether GCQS works well in long-horizon tasks such as AntMaze compared to methods utilizing high-level planning, such as HIGL and PIG.**
>
> Thank you for your suggestion. In the revised version, we have included a comparison with the subgoal-based state-of-the-art method, PIG [1]. The results indicate that our GCQS method can achieve performance comparable to the state-of-the-art method PIG [1], even without the additional algorithmic design for planning or subgoal discovery. Notably, we chose to implement PIG as it demonstrated superior performance over HIGL [2] in the Antmaze environment, as reported in the original PIG study.
>
> **4. It is unclear how GCQS addresses the issue of short trajectory updates since HER already conducts a similar relabeling scheme. Which component of GCQS handles this?**
>
> Existing HER-based methods tend to prioritize shorter segments of trajectories during the update process, meaning that updates often utilize short-length goals from the same trajectory, while longer segments remain unused and this valuable information is completely overlooked. **Our perspective emphasizes leveraging longer trajectory information within the same hindsight framework. To achieve this, we have developed a novel algorithm called GCQS, which can be viewed as an extended version of GCAC. Specifically, GCQS leverages the achieved goals trajectory as the subgoal distribution.**. As illustrated in **Figure 6**, we analyzed the trajectory information utilized by GCQS during the update process. The results indicate that GCQS effectively incorporates trajectory data of varying lengths, showcasing strong performance across different scenarios.
>
> **5. How important is the SAC component in Eq.12 for GCQS? This omitted ablation leaves the question of the importance of the BC term.**
>
> In the ablation study presented in Section 6.3, we examined the influence of Behavior Cloning (BC) on learning to reach achieved goals. This variant, referred to as "No BC-Regularized Q," is depicted by the green curve in Figure 8. The results demonstrate that removing the BC component negatively impacts the performance of GCQS, indicating that incorporating the BC term enhances overall performance.
>
> **6. Presenting more details on the derivation of Eq. 25 from Eq. 24 would be helpful for readers.**
>
> Thanks for the suggestion – we have fixed this.
>
> [1] Kim, Junsu, et al. "Imitating graph-based planning with goal-conditioned policies." arXiv preprint arXiv:2303.11166 (2023).
>
> [2] Kim, Junsu, Younggyo Seo, and Jinwoo Shin. "Landmark-guided subgoal generation in hierarchical reinforcement learning." Advances in neural information processing systems 34 (2021): 28336-28349.

---

> > ### Comment · Reviewer_kpwt · 2024-11-25
> >
> > Thank you for the author's effort in addressing my concerns and clarifying questions. However, I still have some concerns.
> >
> > How long does GCQS take for training compared to existing methods, incorporating explicit subgoal discovery and planning algorithms? Without it, the merits of GCQS simplifying a complex component could be obscure.
> >
> > In addition, the performance of PIG presented in Figure 7 of the revised manuscript is significantly worse than that in the original paper. The result for the L-shaped AntMaze in Figure 7 is also worse than the result of HER in [1]. Could you explain the possible reasons for this performance gap?
> >
> > This discrepancy may be due to insufficient training; however, the x-axis being labeled with 'Epoch' hinders a direct comparison with other papers, which often present performance with respect to environment timesteps.
> >
> > Overall, the questions above raise concerns about the validity of arguing that GCQS achieves performance comparable to other methods that incorporate explicit subgoal discovery and planning.
> >
> > [1] Kim, Junsu, et al. "Imitating graph-based planning with goal-conditioned policies." arXiv preprint arXiv:2303.11166 (2023).

---

> > > ### Author Response · Authors · 2024-12-01
> > > **Author Response**
> > >
> > > We revisited the original paper and code of PIG, and reran the experiments. It was clear that the initial results were due to insufficient training. As a result, we retrained the model and also incorporated several other state-of-the-art subgoal baseline algorithms, such as BEAG [1], DHRL [2], and HIGL [3]. In Section 6.2 of the revised version, the results indicate that GCQS performs slightly worse than BEAG [1] and PIG [4] in most environments, with a small gap to PIG [4]. However, in certain environments, GCQS outperforms DHRL [2] and HIGL [3]. In future work, we believe that designing additional algorithms to identify subgoals from achieved goals presents a promising avenue for further performance improvement, such as through the introduction of graph structures or grids. Additionally, there are several reasons why GCQS outperforms DHRL and HIGL in certain environments.
> > > These reasons include:
> > > - Achieved goals serve as an excellent choice for subgoals.
> > >
> > > - GCQS demonstrates superior performance compared to HIGL for the following reasons: (1) Avoidance of Landmark Dependency: HIGL relies on the selection of fixed landmarks to guide exploration, which can lead to inefficiencies or inconsistencies in landmark selection, especially in dynamic or complex environments. GCQS, on the other hand, leverages achieved goals as subgoals directly within the trajectory, eliminating the need for external landmark selection and offering greater flexibility in adapting to various environments. (2) HIGL may suffer from bias in the selection of landmarks, especially when the landmark distribution is uneven, potentially leading to over-exploration of certain regions. GCQS dynamically generates subgoals from achieved goals, ensuring more balanced exploration and avoiding the bias inherent in fixed landmark-based approaches.
> > >
> > > - GCQS demonstrates superior performance compared to HIGL for the following reasons: The exploration challenge in DHRL lies in its persistent attempt to pursue unattainable subgoals beyond the task boundaries, continuing to succeed only in the vicinity of the initial state, as highlighted in BEAG [1]. Moreover, the graph-based structure of DHRL relies on task decomposition and subgoal discovery. While this approach contributes to refining tasks and enhancing their operability, it also introduces additional complexity, requiring more computational resources to maintain and update the graph structure. This is particularly problematic in long-horizon tasks, where it can increase both the learning time and computational cost. Furthermore, when faced with sparse rewards, DHRL depends on the graph structure to explore and generate subgoals, which may take a longer time to identify effective subgoals. Although the graph structure aids in task decomposition, in sparse reward environments, the generated subtasks may fail to provide timely feedback, slowing down the learning process.
> > >
> > > [1]Yoon, Youngsik, et al. "Breadth-First Exploration on Adaptive Grid for Reinforcement Learning." ICML 2024.
> > >
> > > [2]Lee, Seungjae, et al. "DHRL: a graph-based approach for long-horizon and sparse hierarchical reinforcement learning." NIPS 2022.
> > >
> > > [3]Kim, Junsu, Younggyo Seo, and Jinwoo Shin. "Landmark-guided subgoal generation in hierarchical reinforcement learning." NIPS 2021.
> > >
> > > [4]Kim, Junsu, et al. "Imitating graph-based planning with goal-conditioned policies." ICLR 2023.

---

> > > ### Author Response · Authors · 2024-12-03
> > >
> > > As the author-reviewer discussion phase draws to a close, we eagerly await your feedback regarding whether your previous concerns have been adequately addressed and if there are any remaining questions. In line with your suggestions, the re-experiments and revisions have been incorporated into the updated version, particularly regarding the necessary configurations when training subgoal-based methods, such as ensuring sufficient training duration and labeling the x-axis as the number of environment steps.
> > >
> > > If all concerns have been resolved, we kindly hope you will consider adjusting the rating accordingly.
> > >
> > > Regardless, we wish you all the best!

---

> > > > ### Comment · Reviewer_kpwt · 2024-12-03
> > > >
> > > > Thank you for the authors' effort.
> > > >
> > > > I'm still curious about "How long does GCQS take for training compared to existing methods, incorporating explicit subgoal discovery and planning algorithms? Without it, the merits of GCQS simplifying a complex component could be obscure."
> > > >
> > > > It seems that GCQS does not perform strongly compared to PIG and BEAG in additional benchmark problems such as long-horizon ant maze tasks in Figure 7. What could be the contribution of GCQS replacing complex components in those methods with a simple structure at the expense of performance degradation, considering practicality?

---

> > > > > ### Author Response · Authors · 2024-12-03
> > > > > **Author Response**
> > > > >
> > > > > Thank you for your insightful questions and suggestions.
> > > > >
> > > > > Indeed, training GCQS requires a significant amount of time, and other subgoal-based methods face similar challenges. We conducted training on an RTX 3090 GPU cluster. Based on our calculations, training GCQS on a single GPU with a single seed in a single environment takes approximately 6 hours, while PIG requires between half a day and one day. BEAG is slower than PIG. Roughly estimated, GCQS is 2 times faster than PIG and 3 times faster than BEAG. Additionally, GCQS has the lowest memory usage among the three methods.
> > > > >
> > > > > Regarding the second point, we offer the following perspective:
> > > > >
> > > > > (1) Both PIG and BEAG are based on graph-based subgoal discovery and planning, whereas GCQS, although also relying on subgoals, does not involve any graph-based designs. Therefore, a direct comparison between their performance is not entirely fair, as it fails to properly reflect the strength of our algorithm simply based on relative performance.
> > > > >
> > > > > (2) While the simpler structure of GCQS may lead to some performance degradation compared to methods that utilize more complex techniques, the extensive experiments conducted in this paper (including tasks such as AntMaze, robotics Fetch series, and Hand series) demonstrate that the primary contribution of GCQS lies in its ability to address the challenges identified in Section 4 and its generalizability. The generalizability of GCQS is evident in its capacity to be seamlessly integrated into any HER-based algorithm, yielding superior performance.

---

### Official Review · Reviewer_JyB4 · 2024-11-04

**Soundness:** 3
**Presentation:** 2
**Contribution:** 3
**Rating:** 8
**Confidence:** 4

**Summary:**

This paper studies hindsight relabeling in goal-conditioned reinforcement learning (GCRL). The key finding is that previous GCRL methods have a bias towards closer achieved goals during training, which results in the learned policy being less aligned to reach long-term goals. To address this issue, this paper proposes a novel GCRL method, GCQS, which first learns a goal-reaching policy and then uses the KL-divergence between the learned policy to final goals and goal-reaching policy to subgoals along the trajectories to guide policy optimization for reaching the long-term goals. This method enjoys a performance guarantee and demonstrates better performance than state-of-the-art GCRL baselines.

**Strengths:**

1. The proposed method shows strong empirical results.
2. The method is well-motivated.

**Weaknesses:**

Major questions:

1. If prioritizing the closer achieved goals is the main issue of previous GCRL methods, can we simply use data augmentation to rebalance the relabeling trajectory dataset and improve the performance? I think this would be an interesting result to see and could make the paper more solid.
2. The connection between GCQS and GCWSL: while the authors do not classify GCQS as one GCWSL method, I think there are some similarities to discuss between the two methods. Actually, in offline settings, both Equation 12 and Equation 15 should have a closed-form solution, which is an exponentially weighted form of $\pi_{relabel}$ and $\pi^{prior}$, as discussed in [1]. This makes me wonder if GCQS is still a GCWSL method (like exponential advantage weight mentioned in WGCSL [2]) and if the closed-form solution could further improve the method.
3. Several baselines in the experiments are offline methods, so I am wondering how authors compare them with other online baselines and GCQS to make sure the comparison is fair.

Minor points:

1. The cumulative function should be defined in Theorem 4.1, as it is slightly different from the commonly used probability cumulative function.
2. I believe Section 6.1 has a typo: 16 CPUs -> 16 GPUs.

Despite the questions mentioned above, I am recommending weak acceptance due to the motivation and strong performance of the paper at this stage, however I hope the authors can address my concerns.

[1] Nair, Ashvin, et al. "Awac: Accelerating online reinforcement learning with offline datasets." *arXiv preprint arXiv:2006.09359* (2020).

[2] Yang, Rui, et al. "Rethinking goal-conditioned supervised learning and its connection to offline rl." *arXiv preprint arXiv:2202.04478* (2022).

**Questions:**

There are some questions and concerns, which I have outlined in the previous section.

---

> ### Author Response · Authors · 2024-11-20
> **Author Response**
>
> We thank the reviewer for the positive feedback!
>
> **1. If prioritizing the closer achieved goals is the main issue of previous GCRL methods, can we simply use data augmentation to rebalance the relabeling trajectory dataset and improve the performance? I think this would be an interesting result to see and could make the paper more solid.**
>
> Thank you very much for presenting this interesting idea. However, I am not entirely clear on how data augmentation would be specifically implemented in this context. Could you please provide a more detailed explanation, or suggest relevant literature for further study? I look forward to your feedback.
>
> **2. The connection between GCQS and GCWSL: while the authors do not classify GCQS as one GCWSL method, I think there are some similarities to discuss between the two methods. Actually, in offline settings, both Equation 12 and Equation 15 should have a closed-form solution, which is an exponentially weighted form of $\pi_relabel$ and $\pi^{piror}$, as discussed in [1]. This makes me wonder if GCQS is still a GCWSL method (like exponential advantage weight mentioned in WGCSL [2]) and if the closed-form solution could further improve the method.**
>
> In the offline setting, the closed-form solutions for Equation (12) and Equation (15) align with GCWSL. However, in the online setting, we directly optimize the Q-function and the behavior cloning (BC) objective, positioning GCQS as an extension of GCAC.
>
> **3. Several baselines in the experiments are offline methods, so I am wondering how authors compare them with other online baselines and GCQS to make sure the comparison is fair.**
>
> Firstly, the WGCSL [1] paper indicates (i.e., in line 14 of the abstract and the results in Figure 15) that WGCSL is applicable to both online and offline settings. Secondly, [2] indicates that WGSL and GoFar belong to the family of Advantage-weighted Regression (AWR). Finally, the work in [3] also highlights the relationship between DWSL and AWR. Given that AWR [4] is effective in online settings, we believe these baselines are also applicable in the online setting. Additionally, ActionModels [5] combines goal-chaining with conservative Q-Learning, which, as noted, is not suitable for the offline setting. Consequently, we have removed this baseline in the revised version. In the revised version, we have included a comparison with the subgoal-based state-of-the-art method, PIG [6]. The results indicate that our GCQS method can achieve performance comparable to the state-of-the-art method PIG [6], even without the additional algorithmic design for planning or subgoal discovery.
>
> **4. The cumulative function should be defined in Theorem 4.1, as it is slightly different from the commonly used probability cumulative function.**
>
> Thanks for the suggestions here. This has been addressed in the revised version of the paper.
>
> **5. I believe Section 6.1 has a typo: 16 CPUs -> 16 GPUs.**
>
> Thanks for the suggestion – we have fixed this.
>
> [1] Yang, R., Lu, Y., Li, W., Sun, H., Fang, M., Du, Y., … Zhang, C. (2022). Rethinking Goal-conditioned Supervised Learning and Its Connection to Offline RL. ICLR.
>
> [2] Yang, Wenyan, et al. "Swapped goal-conditioned offline reinforcement learning." arXiv preprint arXiv:2302.08865 (2023).
>
> [3] Hejna, Joey, Jensen Gao, and Dorsa Sadigh. "Distance weighted supervised learning for offline interaction data." International Conference on Machine Learning. PMLR, 2023.
>
> [4] Peng, Xue Bin, et al. "Advantage-weighted regression: Simple and scalable off-policy reinforcement learning." arXiv preprint arXiv:1910.00177 (2019).
>
> [5] Chebotar, Yevgen, et al. "Actionable models: Unsupervised offline reinforcement learning of robotic skills." arXiv preprint arXiv:2104.07749 (2021).
>
> [6] Kim, Junsu, et al. "Imitating graph-based planning with goal-conditioned policies." arXiv preprint arXiv:2303.11166 (2023).

---

> > ### Comment · Reviewer_JyB4 · 2024-11-22
> >
> > Thank you for the clarification.
> >
> > As for point 1, I am thinking of upsampling longer trajectories in the training data to balance the unbalanced distribution (as depicted in Figure 2). One very naive example could be sampling the data with the weight 1/H, where H is the trajectory length, also I am sure that there will be clever ways to apply this idea.

---

> ### Author Response · Authors · 2024-12-01
> **Author Response**
>
> We appreciate your simple yet insightful suggestion. Following your approach, we implemented weighted sampling, which favors longer trajectories. The results are presented in Appendix D.1 of the revised version [1]. Our findings show that while simple weighted sampling (or upsampling) does improve the performance of the baseline algorithms, it still does not outperform GCQS. This underscores the significance of the subgoal mechanism in GCQS, which enables the learning of more optimal actions.
>
> [1] https://pdfhost.io/v/~QVHMgEkR_ICLR2025_GCQS_Rebuttal

---

> > ### Comment · Reviewer_JyB4 · 2024-12-01
> > **Thanks for the response**
> >
> > Thank you for providing additional results for my question. I have raised my score since the author has addressed most of my concerns.

---

> > > ### Author Response · Authors · 2024-12-02
> > > **Author Response**
> > >
> > > Once again, we express our sincere gratitude for the affirmation of our work and the valuable feedback you have provided!

---

### Author Response · Authors · 2024-12-04
**Response to all reviewers**

We thank all reviewers for their constructive feedback! We included new experimental results and revisions in the PDF following the reviewers’ suggestions. Please check out our revised paper. We highlighted the changes in the revised version.

Here, we summarize the major changes:

- Revised the paper to correct unreasonable grammar and inappropriate expressions.

- Added a comparison with such subgoal-based state-of-the-art algorithms, such as HIGL [1], DHRL [2], PIG [3] and BEAG [4]  (Section 6.2). The results from complex, long-horizon Antmaze experiments indicate that GCQS demonstrates performance that is slightly inferior to or comparable with PIG, while outperforming both HIGL and DHRL. Additionally, GCQS achieves twice the training speed of PIG and three times that of BEAG, with significantly reduced memory costs. Extensive experiments across diverse tasks—including Antmaze, the Robotics Fetch series, and the Hand manipulation tasks—highlight the primary contributions of GCQS: addressing the challenges discussed in Section 4 and offering broad generalizability. Notably, the generalizability of GCQS lies in its ability to integrate seamlessly into any HER-based algorithm, delivering superior performance enhancements.

- Included error bar plots in Figure 6 to enhance the visual representation of the results. (Appendix Section D.4)

- Removed the offline baseline “action models” and added detailed explanations for baselines such as WGCSL, GoFar, and DWSL. These changes do not affect the results or main conclusions of the paper. (Figure 6 and Appendix C.1)

- The primary motivation of GCQS has been clarified as effectively leveraging longer trajectory information within the same hindsight framework (Figure 11).

- An ablation study on the non-uniform sampling of longer trajectories, such as weighted sampling or upsampling, for achieved goals (Appendix D.1).

Overall, our new findings and the novel extension framework based on GCAC, GCQS, represent an innovative and interesting contribution to goal-conditioned reinforcement learning (GCRL). Overall, our new findings and the novel extension framework based on GCAC and GCQS represent a significant and innovative contribution to goal-conditioned reinforcement learning (GCRL). The GCQS framework, which is both simple and highly efficient, not only achieves state-of-the-art performance and improved robustness on challenging goal-conditioned benchmarks, but also demonstrates competitive results on the long-horizon AntMaze task, comparable to some advanced subgoal-based methods. The final revised version of the paper can be found in https://pdfhost.io/v/~QVHMgEkR_ICLR2025_GCQS_Rebuttal.

[1]Kim, Junsu, Younggyo Seo, and Jinwoo Shin. "Landmark-guided subgoal generation in hierarchical reinforcement learning." NIPS 2021.

[2]Lee, Seungjae, et al. "DHRL: a graph-based approach for long-horizon and sparse hierarchical reinforcement learning." NIPS 2022.

[3]Kim, Junsu, et al. "Imitating graph-based planning with goal-conditioned policies." ICLR 2023.

[4]Yoon, Youngsik, et al. "Breadth-First Exploration on Adaptive Grid for Reinforcement Learning." ICML 2024.

---

### Meta-Review · Area_Chair_xgtx · 2024-12-21

**Metareview:**

This paper studies a tendency in existing GCRL methods with goal relabelling to prefer optimizing towards closer goals. To mitigate this bias, the paper regularizes the policy to take similar actions for the final desired goal and intermediate subgoals. Experiments demonstrate that the proposed method achieves strong performance with fewer environment interactions.

Reviewers appreciated the clear motivation/writing/figures and strong empirical results. Some reviewers appreciated the theoretical support for the paper.

Multiple reviewers suggested comparing to a baseline that rebalanced the training data to skew the distribution over short-horizon vs long-horizon goals. Reviewers had questions about the relationship with some prior work (e.g., GCWSL, PIG, performance on FetchSlide), and recommended including additional baselines designed for the online setting. Reviewers had a few questions about the paper presentations, such as how results were smoothed and averaged across seeds, whether there is an inherent tradeoff in prioritizing short vs long horizon tasks.

Taken together, my recommendation is that the paper be rejected.

**Additional Comments On Reviewer Discussion:**

In a discussion among the reviewers, the consensus was that the paper makes some interesting contributions yet several concerns remain unresolved (e.g., method complexity).

The authors argued that the comparison with some of the suggested baselines (PIG, BEAG) would be unfair; I disagree with this characterization: while these methods are structurally distinct from the proposed method, I think it would be possible to do a fair comparison as I think these graph-based methods make the same assumptions as the proposed method. The authors report that the proposed method performs worse than PIG.

---

### Decision · Program_Chairs · 2025-01-22

Reject